

# The impact of calibration strategies on future evapotranspiration projections: a SWAT-T comparison of three hydrological modeling approaches in West Africa

Fabian Merk[1], Timo Schaffhauser[1], Faizan Anwar[1], Manuel Rauch[2], Jan Bliefernicht[2], and Markus Disse[1]

[1]School of Engineering and Design, Technical University of Munich, Munich. Germany
[2]Institute of Geography, University of Augsburg, Augsburg. Germany

**Correspondence:** Fabian Merk (fabian.merk@tum.de)

**Abstract.**

Actual evapotranspiration (AET) is pivotal for the assessment of current and future water availability, particularly for sub humid and AET dominant regions such as West Africa. In this region, climate change is projected to be substantial, which will catalyze hydrological changes. In the climate-hydrological modeling chain for impact assessment, multiple sources of uncer-

tainty are embedded. While the uncertainties inherent in general circulation models (GCM) are difficult to reduce, minimizing uncertainties from hydrological modeling remains a critical focus for researchers and practitioners. Hence, the present study investigates the impact calibration strategies can have on future hydrological changes in West Africa. Given the key role of AET in West Africa, the study particularly evaluates how calibration shapes its future dynamics. In addition, we test whether a specific plant growth modeling, attributed as leaf area index (LAI), can be used as a proxy to predict AET. The Bétérou Catch-

ment in Benin is selected as a demonstration case along hydrological modeling with the eco-hydrological SWAT-T model. To investigate calibration impacts, we apply three strategies, which range from simple (discharge (Q) only) to more comprehensive (Q and LAI; Q, LAI, and AET) approaches. We use the Robust Parameter Estimation algorithm in each calibration strategy to address parameter equinfinality. We use the standardized future climate data from ISIMIP3b (CMIP6) with five GCMs and three emission scenarios and evaluate changes for the near (2031–2050) and far (2070–2099) future periods. The findings show

that the amount of future annual AET depends on the calibration strategy, where the change signal for all strategies indicates AET increases. The approach including AET calibration (Q, LAI, AET) shows high future changes, with e.g., multi-model mean changes for SSP5–8.5 of $\Delta E_{near} = 5.8\%$ and $\Delta E_{far} = 8.4\%$. The results moreover demonstrate that the combined "Q + LAI" can be used as a proxy to predict AET rates. For discharge, the change signal mostly indicates future decreases across all calibration strategies with multi-model mean changes for SSP5–8.5 of $\Delta Q_{far} = -7.0\%$ (Q, LAI, AET) to $\Delta Q_{far} = -1.6\%$

(Q only). Yet, contrasting predictions of future changes depending on single GCMs are simulated. The present study under-scores the relevance of uncertainty integration in climate-hydrological modeling and contributes to an improved understanding of water availability assessment in West Africa.





## 1 Introduction

Actual evapotranspiration (AET) is a key hydrological attribute to determine water availability (Chawanda et al., 2024), and to
estimate agricultural, vegetation, and drought stressors (Rind et al., 1990; Fisher et al., 2017; Miralles et al., 2025). It plays an essential role in the regional hydrology in the sub-humid parts of West Africa, where the rate of AET to precipitation can be up to 80 % (Rodell et al., 2015). In tropical and perennial vegetated regions, the contribution of plant transpiration to AET is particular high (Wei et al., 2017) given the direct linking of plant growth and leaf area index (LAI) to the canopy conductance (Good et al., 2014; Wang et al., 2014).

Climate change impacts on the future hydrology in West Africa are projected to be substantial. It is reasonably certain that precipitation and temperature intensities increase for the entire West African region, specifically leading to higher mean air temperatures, more frequent extreme heat days, and recurring sub-daily heavy precipitation events (IPCC, 2022). The change of mean annual precipitation in West Africa is spatially contrasting, where drying in the western and wettening in the eastern parts are projected (IPCC, 2022). The hydrological response to future climate changes is typically assessed in a climate-
hydrological modeling chain. In this, hydrological (impact) models are forced with downscaled climate projections of general circulation models (GCMs) under various greenhouse gas emission scenarios. Since hydrological responses to future climate conditions are assessed using models calibrated under current climate, robust calibration strategies and model parametrisation are essential (Krysanova et al., 2017; Hattermann et al., 2018).

The sensitivity of the calibration strategy on the regional-scaled climate impact assessment is well-documented (Huang
et al., 2020; Ismail et al., 2020; Koch et al., 2020; Mishra et al., 2020; Wen et al., 2020). These studies report impacts of simple (discharge only) compared to comprehensive (multi-gauge or multi-objective) calibration strategies, particularly with respect to future discharge changes. For instance, future projected discharge can be up to 10 % higher for models from multi-objective calibration compared to discharge only approaches (Huang et al., 2020).

These studies however relied on single-best model parametrisations for the climate impact assessment. Model equifinality
impairs robust hydrological modeling for current (Beven, 2006) and future climate (Her et al., 2019), because multiple model parameter sets can accurately represent the hydrological cycle. Different approaches to address parameter equifinality exist, such as Generalized Likelihood Uncertainty Estimation (Beven and Binley, 2014), Robust Parameter Estimation (ROPE) (Bárdossy and Singh, 2008), or Markov Chain Monte Carlo methods (Roberts and Rosenthal, 2004), among others, with which multiple equally-well performing parameter sets can be derived. The uncertainty contributions from GCMs typically exceed
those stemming from parameter equifinality, as shown by studies such as Her et al. (2019) or Brigode et al. (2013). Yet, the direction and magnitude of future hydrological changes can still vary considerably depending on the calibration strategy and parameter sets (Mendoza et al., 2016).

Among other hydrological impact models, the Soil and Water Assessment Tool (SWAT) model and its open-source successor SWAT+ have been widely applied for climate impact assessment in West Africa (Akoko et al., 2021). It is a process-describing
model specified to represent catchment hydrology, crop management and vegetation growth (Arnold et al., 1998). Climate impact studies have covered the Niger Basin (Angelina et al., 2015; Eisner et al., 2017; Krysanova et al., 2017; Animashaun



et al., 2023; Chawanda et al., 2024); the Mono River (Houngue et al., 2023); the Ouémé Basin (Bossa et al., 2012, 2014; Danvi et al., 2018); the Volta (Kankam-Yeboah et al., 2013; Sood et al., 2013), Pra (Awotwi et al., 2021), Owabi (Osei et al., 2019), and Vea River (Larbi et al., 2021) in Ghana; and the Tougou River (Yonaba et al., 2023). The studies indicate that precipitation
projections for West Africa are variable depending on the applied GCM, emission scenarios and future period. This variability translates into varying streamflow projections across the region, where the Niger Basin indicates mixed discharged trends, the Ouémé Catchment reports decreased discharges, and streamflow increases for the Mono River Catchment.

These studies mostly focus their modeling on discharge, and AET – despite being a key driver of the regional hydrological cycle – has been integrated in the model evaluation only partially. Few studies, such as Chawanda et al. (2024), calibrated
the SWAT+ model against discharge and AET. Most others estimate future AET from SWAT models being optimized with discharge-only calibration approaches (Bossa et al., 2012; Sood et al., 2013; Bossa et al., 2014; Danvi et al., 2018; Larbi et al., 2021; Animashaun et al., 2023). However, the modeling of AET in process-describing hydrological models, such as SWAT/SWAT+, substantially relies on parameters that also influence runoff generation, e.g., interception or soil moisture. Hence, without explicit model calibration against AET, projections of future AET can remain constrained and potentially
misleading.

In this study, we explain how calibration strategies with hydrological models can influence the simulation outcomes. We demonstrate how future rates of AET and discharge can differ if or if not AET is considered in the model calibration. In addition, this study evaluates the role of vegetation (attributed as LAI) in assessing AET. We test whether LAI can be used as a proxy to predict future AET dynamics. To date, no study has yet discussed the implications of AET being disregarded in the
calibration but used for future assessments in West Africa.

We evaluate the implication of three different model calibration strategies on the climate impact assessment. The strategies include: i) a conventional single-objective optimization using only streamflow at the catchment outlet ("Q only"); ii) a multi-objective optimization considering streamflow at the outlet and LAI at the subbasin scale ("Q + LAI"); and iii) a multi-objective optimization integrating streamflow at the outlet, along with LAI and AET at the subcatchment scale ("Q + LAI
+ AET"). We derive multiple equally-well performing parameter sets with the ROPE algorithm to address the uncertainty from equifinality. ROPE is applied for single- (discharge) and multi-objective optimisation, where discharge, AET, and further information like the vegetation attribute LAI are considered. These models are used for the climate impact assessment to investigate the difference between single-best and multiple near-optimal parameter estimations, particularly regarding AET. As a demonstration case, we apply the SWAT-T (for tropics) model for the Bétérou Catchment in Benin.

This study contributes to minimize uncertainties from model calibration approaches in the climate-hydrological model coupling. We make use of statistically downscaled climate model data from the Inter-Sectoral Impact Model Intercomparison Project (ISIMIP) that provides standardized climate impact data for cross-sectoral studies. We force the hydrological model with climate data from ISIMIP3b (five GCMs and three emission scenarios) based on the Coupled Model Intercomparison Project Phase 6 (CMIP6) (Lange, 2019) to minimize uncertainties from downscaling, bias adjustment, and scenario framing.



## 2  Methods

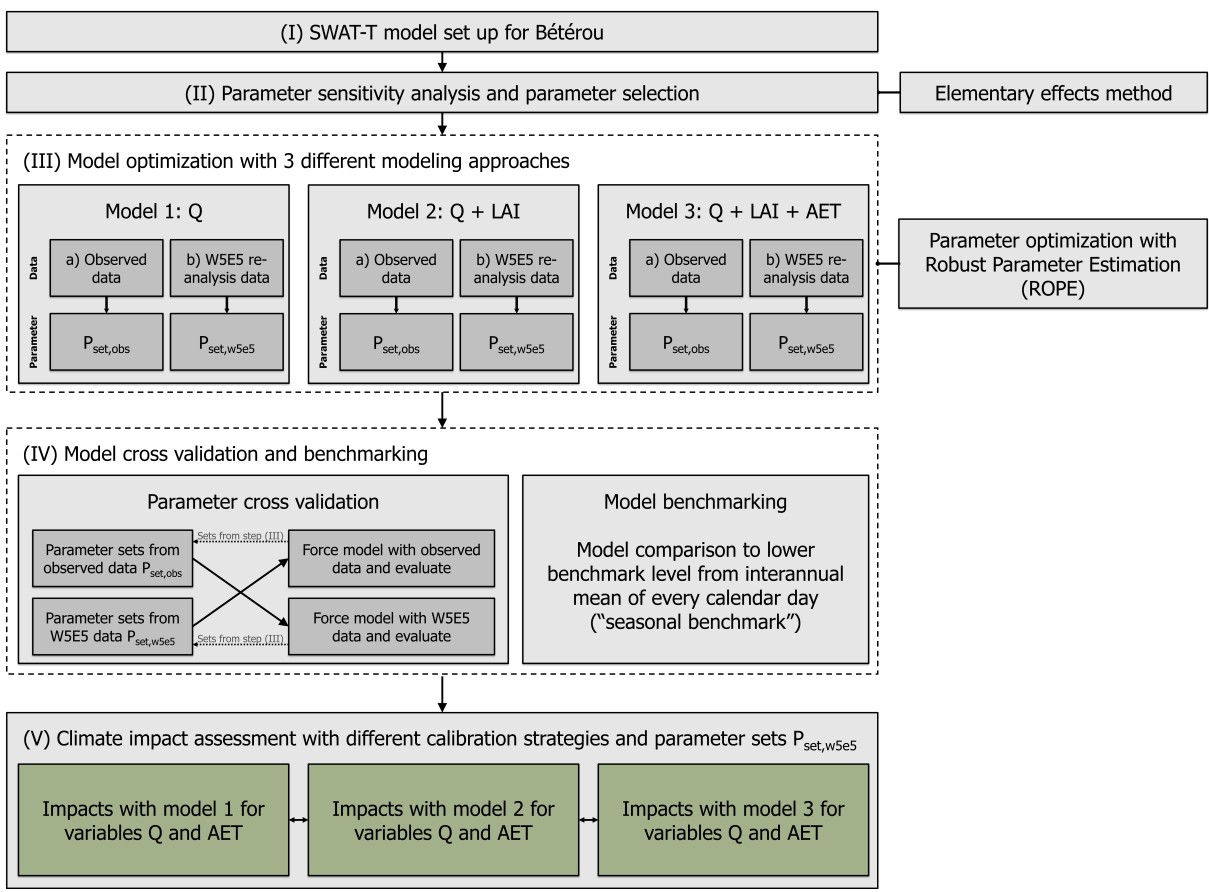

**Figure 1.** Methods applied in the present study to evaluate different calibration strategies for climate impact assessment.

Figure 1 gives an overview of the methods applied in this study. First, the basin-scale SWAT-T model is set up for the Bétérou Catchment in Benin. The parameter sensitivity is quantified with the elementary effects method after Morris (Morris, 1991), and, subsequently, relevant parameters are selected. We apply the Morris method, because its computation of the elementary effects allows to quantify and rank non-linear model responses, and identify (non-) influential parameters (Morris, 1991;

Campolongo et al., 2007). Three different calibration strategies are investigated: model optimization regarding discharge only; discharge and LAI; and discharge, LAI, and AET. We use the ROPE algorithm (Bárdossy and Singh, 2008) to derive multiple well-performing parameter sets and to address parameter equifinality. We force the hydrological model with observed (Galle et al., 2018; Schlueter et al., 2019; Vogel et al., 2018; Rauch et al., 2024) and the W5E5 meteorological data (Lange et al., 2021). The robustness of the model optimization is checked with a cross validation and model benchmarking. Finally, the

role of the calibration strategy is evaluated with respect to the climate impact assessment, where future hydrological response



(discharge and AET) predicted by each of the three approaches is evaluated. The responses for near and far future climate projections are derived and the implications of different modeling approaches are discussed.

## 2.1 Model description and parameter selection

The SWAT-T model is a further development of the SWAT ecohydrological model (Arnold et al., 1998). The key enhancement

of SWAT-T is the modification of the plant growth module to better represent the plant phenology in tropical regions (Alemayehu et al., 2017). The SWAT-T model leads to more accurate predictions of AET in regions with perennial plant growth (Zhang et al., 2020; Fernandez-Palomino et al., 2021; Ferreira et al., 2021; López López et al., 2017; Merk et al., 2024). Aside from the plant module modification, SWAT-T remains identical to SWAT. The original SWAT model has been applied for river basins worldwide (Arnold and Fohrer, 2005; Tan et al., 2020), as well as regionally in West Africa and for different catchments

in Benin (Akoko et al., 2021). In Benin, most studies focused on the discharge simulation for the Ouémé River basin (Bossa et al., 2014; Poméon et al., 2018) and as its subcatchments (Giertz et al., 2006; Bossa et al., 2012; Duku et al., 2016, 2018; Danvi et al., 2017; Togbévi et al., 2020). Remotely sensed AET has also been a primary model optimization target to predict streamflow (Odusanya et al., 2019, 2021). To the best of our knowledge, the SWAT-T model has only been applied in Benin for the assessment of the LAI–AET interaction (Merk et al., 2024), which demonstrates the AET prediction through LAI on

the footprint scale.

The SWAT/SWAT-T model is a process-oriented ecohydrological model for the catchment-scale water balance estimation. It partitions the computation of the water balance into five storages: snow, canopy storage, soil profiles, and a shallow and a deep aquifer. The water balance in SWAT/SWAT-T is described as:

$$\Delta S = \sum_{i=1}^{N} \left( P - Q_{total} - E - w_{losses} \right), \tag{1}$$

where $\Delta S$ is the change in storage; $N$ is the time in days or months; $P$ is the precipitation; $Q_{total}$ is the total runoff (in SWAT "water yield") as a sum of surface runoff, lateral flow, and base flow; $E$ is AET; and $w_{losses}$ are deep groundwater losses. We use the Soil Conservation Service (SCS) curve number (CN) method to model the surface runoff. The SWAT/SWAT-T model is spatially discretized into subbasins and further subdivided into hydrological response units (HRUs). Generally, three methods are available to compute potential ET ($E_0$) in SWAT/SWAT-T: the Hargreaves method (Hargreaves and Samani,

1985), the Priestley-Taylor method (Priestley and Taylor, 1972), and the Penman-Monteith method (Monteith, 1965). Once $E_0$ is calculated, it is partitioned into potential plant transpiration ($T_{plant}$) and potential soil evaporation ($E_{soil}$). We use the Penman-Monteith method since it integrates temperature- and energy-based assumptions and vegetation. Merk et al. (2024) demonstrated that this method outperforms the other two options for AET prediction for characteristic West African land cover types (forest and shrubland), which are also most dominant in the Bétérou Catchment. $E_0$ after Penman-Monteith in

SWAT/SWAT-T is computed as:

$$E_0 = \frac{\Delta \cdot (R_{net} - G) + \rho_{air} \cdot c_p \cdot (e_z^0 - e_z)/r_a}{\lambda \cdot (\Delta + \gamma \cdot (1 + r_c/r_a))}, \tag{2}$$





where $\lambda$ is the latent heat of vaporization; $R_{net}$ is the net radiation; $\Delta$ is the slope of the saturation vapor pressure-temperature curve; $\rho_{air}$ is the air density; $c_p$ is the specific heat at constant pressure; $e_z^0$ is the saturation vapor pressure of air at height $z$; $e_z$ is the water vapor pressure of air at height $z$; $r_a$ is the aerodynamic resistance; and $r_c$ is the plant canopy resistance.

In SWAT/SWAT-T with $E_0$ after Penman-Monteith, $r_a$ and $r_c$ are attributed to an alfalfa grass reference with constant values (Neitsch et al., 2011). For the computation of $T_{plant}$, the Penman-Monteith equation is also used, and $r_a$ and $r_c$ are derived from an actual plant, where the plant's modeled canopy height and LAI are considered. $E_{soil}$ is then determined as $E_{soil} = E_0 - T_{plant}$. Actual plant transpiration and soil evaporation are computed based on water availability and biophysical factors, such as LAI, root depth, and soil properties such as field capacity. The total actual evapotranspiration (AET) is the sum of

actual plant transpiration and soil evaporation. We use the Penman-Monteith method for the computation of $E_0$. For brevity, a detailed explanation of the plant growth module and LAI computation with SWAT-T is given in the supplementary material to this study.

A total of 26 parameters have been investigated in this study (Table 1). The list of 26 parameters is gathered with a literature review, where selected SWAT studies in Benin (Schuol and Abbaspour, 2006; Schuol et al., 2008; Bossa et al., 2012, 2014;

Duku et al., 2016; Danvi et al., 2017; Poméon et al., 2018; Hounkpè et al., 2019; Togbévi et al., 2020; Odusanya et al., 2019, 2021) and SWAT-T studies (Alemayehu et al., 2017; López-Ramírez et al., 2021; Fernandez-Palomino et al., 2021; Ferreira et al., 2021; Merk et al., 2024) have been screened. The focus was on peer-review studies, where the parameter choice and calibration strategy is transparent. All 26 parameters in Table 1 are evaluated in the sensitivity analysis. Based on the sensitivity ranking, the 15 most influential parameters are considered for the model optimization. The plant parameters FRGRW$_2$,

LAIMX$_2$, DLAI, T_BASE, PHU, and GSI are specified in SWAT-T for each land cover type. The Bétérou Catchment is primarily covered by forest, shrubland, and cropland. To avoid an excessive number of parameters (e.g., 5 plant parameters × 3 land cover types), we transfer the parameter values for FRGRW$_2$, LAIMX$_2$, DLAI, T_BASE, PHU, and GSI for each land cover type from Merk et al. (2024), who investigated specific sites within the Bétérou Catchment. In the model optimization, each plant parameter is changed through a multiplier from 0.75 to 1.25 (see Tab. A4).

Insensitive parameters are not considered in the optimization but kept constant (Table 1). The parameters ALAI_MIN and BLAI are derived based on the maximal and minimal values of the GLASS-LAI time series, respectively. The multipliers for the plant parameters and constant values for insensitive parameters are used in the optimization to reduce the parameter space and address parameter equifinality.

## 2.2 Study site and model set up

The Bétérou catchment is the Northern most headwater catchment of the Ouémé River Basin in Benin (Figure 2). It covers an area of 10100 km$^2$. The catchment is relatively flat with an altitude range from 248 m to 598 m above sea level. The land cover is a typical West African combination mainly consisting of gallery forests (52 %), savannah (33 %), and agriculture (15 %), where the landscape is marked by scattered mosaics of woody savannah with grassy understories. The region is in a natural state with a small share of urban areas (<1 %) (Judex and Thamm, 2008).





**Table 1.** List of parameters used with their description and a cross ("X") if the parameter is considered in the optimization in this study. The superscripts denote the SWAT-T/West Africa study the parameter has been investigated: [1] to [16] correspond to Schuol and Abbaspour (2006); Schuol et al. (2008); Bossa et al. (2012, 2014); Duku et al. (2016); Danvi et al. (2017); Poméon et al. (2018); Hounkpè et al. (2019); Togbévi et al. (2020); Odusanya et al. (2019, 2021); Alemayehu et al. (2017); López-Ramírez et al. (2021); Fernandez-Palomino et al. (2021); Ferreira et al. (2021); Merk et al. (2024), respectively. The table also lists the studies from which the parameters for the less sensitive parameter are transferred from.

| Parameter | Description (unit) | Optimization |
|---|---|---|
| Parameters identified in the sensitivity analysis to be sensitive | | |
| FRGRW$_2$[12,13,14,16] | Fraction of PHU as second point on the optimal LAI curve (–) | X |
| DLAI[12,13,14,16] | Fraction of total PHU when leaf area begins to decline (–) | X |
| T_BASE[12,13,14,16] | Minimum temperature for plant growth (°C) | X |
| PHU[12,14,16] | Total number of heat units needed for plant maturity (–) | X |
| GSI[10,12,16] | Maximum stomatal conductance (m s$^{-1}$) | X |
| CAN_MX[10,11,15,16] | Maximum canopy storage (mm) | X |
| ESCO[1,2,3,5,6,7,8,9,10,11,12,13,15,16] | Soil evaporation compensation factor (–) | X |
| EPCO[5,6,7,8,10,11,12,13,15,16] | Plant uptake compensation factor (–) | X |
| SOL_AWC[1,2,3,4,5,6,7,8,9,10,11,12,13,14,16] | Available water capacity of the soil layer (mm) | X |
| SOL_BD[2,7,8,11,14,16] | Moist bulk density (g cm$^{-3}$) | X |
| GW_REVAP[2,3,5,6,7,8,9,12,15,16] | Groundwater re-evaporation coefficient (–) | X |
| GWQMN[1,2,3,4,5,6,7,8,9,12,15] | Threshold depth of water for baseflow to occur (mm) | X |
| RCHRG_DP[1,2,3,7,8,9,12,14,15,16] | Deep aquifer percolation fraction (–) | X |
| REVAPMN[1,2,3,5,7,8,9,11,12,15,16] | Threshold depth of water for re-evaporation to occur (mm) | X |
| GW_DELAY[2,3,4,6,7,8,9,11,13,14,15] | Ground water delay (day) | X |
| Parameters identified in the sensitivity analysis to be less sensitive | | |
| BLAI[12,13,14,16] | Maximum potential leaf area index (m$^2$ m$^{-2}$) | Observed data |
| ALAI_MIN[12,13,14,16] | Minimum LAI for plant during dormant period (m$^2$ m$^{-2}$) | Observed data |
| SOL_K[3,4,7,8,9,10,13,14,16] | Saturated hydraulic conductivity (mm hr$^{-1}$) | Judex and Thamm (2008) |
| FRGRW$_1$[12,13,14,16] | Fraction of PHU as first point on the optimal LAI curve (–) | Merk et al. (2024) |
| LAIMX$_1$[12,13,14,16] | Fraction of BLAI as first point on the optimal LAI curve (–) | Merk et al. (2024) |
| LAIMX$_2$[12,13,14,16] | Fraction of BLAI as second point on the optimal LAI curve (–) | Merk et al. (2024) |
| T_OPT[12,14,16] | Optimal temperature for plant growth (°C) | Merk et al. (2024) |
| SURLAG[1,2,4,5,7,8,9,12,14,15] | Surface runoff lag coefficient (–) | Bossa et al. (2014) |
| SOL_RD[16] | Maximum rooting depth of soil profile (mm) | Merk et al. (2024) |
| ALPHA_BF[2,3,4,6,8,9,10,11,12,13,15] | Base flow alpha factor (day$^{-1}$) | Bossa et al. (2014) |
| CN2[1,2,3,4,5,6,7,8,9,10,11,12,13,14,16] | Initial SCS runoff curve number (–) | Alemayehu et al. (2017) |





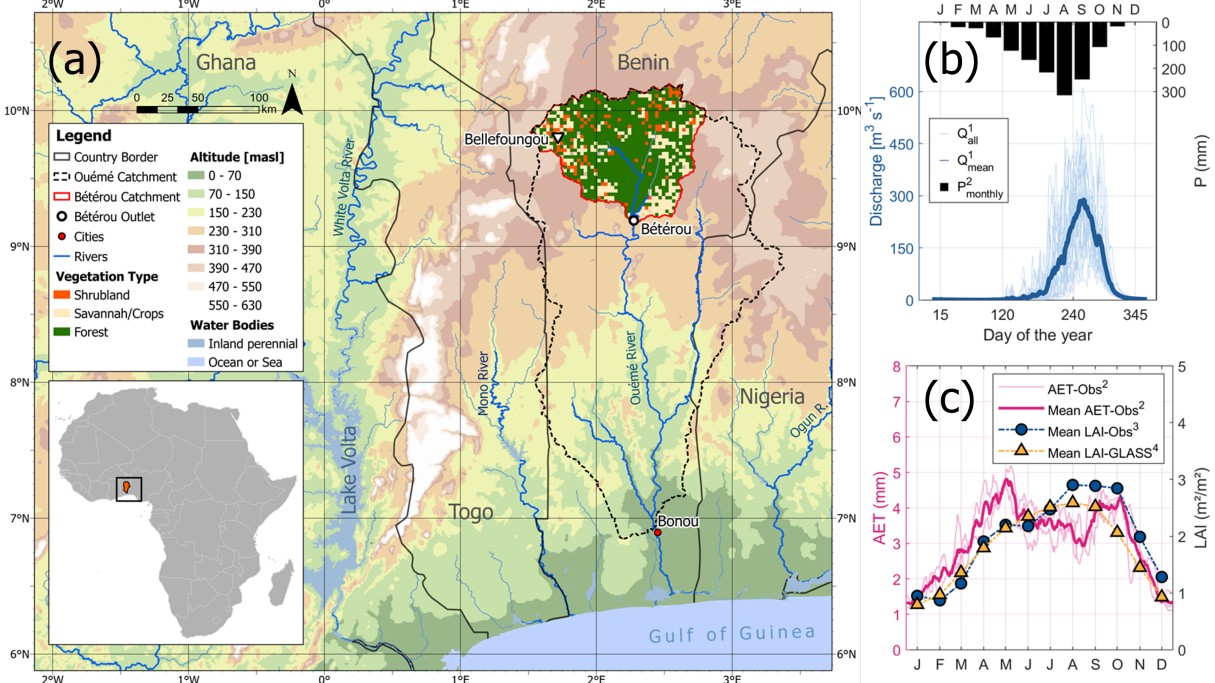

**Figure 2.** (a) Overview of the Bétérou Catchment; (b) monthly mean values of precipitation (Bellefoungou) and daily mean values of discharge (Bétérou); (c) and measurements of AET and LAI in Bellefoungou: an overview of observed AET dynamics (in purple), mean monthly observed LAI (in blue) as well as the mean monthly values of GLASS LAI (in yellow). References for the data used to display the map are also listed in the Appendix (see Table A6). The superscripts in panels (b) and (c) denote the corresponding time periods: [1] January 1996 to December 2020; [2] in (b) January 2006 to December 2020; [2] in (c) January 2008 to December 2010; [3] from July 2008 to May 2010; [4] from January 2007 to December 2015. Publisher's remark: please note that the above figure contains disputed territories.

The agriculture in this region is combination of cash crops, such as cotton, and subsistence-driven smallholders. The cultivated crops are usually crops like maize, yam, sorghum, bean, millet, or cassava (Janssens et al., 2010). The cropping calender in this region follows the rainy season and can generally be dated from May to November – with crop specific variations in planting and harvesting (Forkuor et al., 2014). The distinct dry and wet streamflow seasonality is typical for catchments along the Guinean Savanna. The maximum daily average streamflow occurs in September with an estimated value of 290 m$^3$ s$^{-1}$. The

climate in the Bétérou catchment is typical for Sub-Saharan, sub-humid Africa. The annual precipitation ranges from 1100 to 1500 mm (Mamadou et al., 2016; Bliefernicht et al., 2019). The precipitation pattern is unimodal, with a rainy season between April and October, whereas from November to March, the dry season occurs. The annual mean daily temperature is 25 °C (Galle et al., 2018). The soils in the Bétérou catchment consist of ferric soils with loamy sand present in the upper soil horizons (Giertz and Diekkrüger, 2003). Generally, the AET data follows the seasonal pattern of LAI (Fig. 2c). In addition, a decrease

in AET in the wet season can be observed for this region. During this season, the atmospheric water demand is reduced due high air humidity resulting to low water pressure deficits as shown by Mamadou et al. (2016).





For the model set up, we subdivided the Bétérou Catchment into 14 subbasins and 126 HRUs. The dominant land cover types forest, shrubland, and cropland are assigned to the SWAT/SWAT-T data base of "FRSD", "RNGE", and "AGRL", respectively. The share of cropland cover is relatively small (15 %) and data for detailed crop modeling is scarce. Thus, we apply a conceptual

approach to integrate crops. We assign the generic plant data base "AGRL". For cropland, the plant growth is not automatically triggered by the soil moisture index but its planting and harvesting is scheduled in the management operations to follow the rainy season (Forkuor et al., 2014). We validate the conceptual approach by comparing the simulated LAI of croplands to the reference of GLASS-LAI. The simulations are run for daily timesteps.

## 2.3 Data used in this study

The SWAT-T model is set up using the data listed in Table 2. We force the models with daily data for precipitation, temperature, relative humidity, solar radiation, and wind speed from two data sets. The first set of forcing data is the ISIMIP3b baseline data set from ISIMIP3b (W5E5 data, Lange et al. (2021)). Secondly, we use observed data (AMMA-CATCH (Galle et al., 2018) and precipitation data from KASS-D (Schlueter et al., 2019; Vogel et al., 2018)). The precipitation point observations were regionalized to a 0.05° grid ( 5 km resolution) using a simple nearest-neighbor interpolation approach (Rauch et al., 2024). The

resulting gridded rainfall fields were then spatially averaged over each sub-catchment to produce 14 sub-catchment-specific rainfall time series, which served as precipitation inputs in the model setup. Both data sets (W5E5 and observed data) are used for a separate calibration.

As a reference to SWAT-T's simulated LAI, we use the GLASS-LAI data, because it is reasonable performance in applications worldwide (Liang et al., 2014). Typically, raw LAI data from satellite-based products (e.g., MODIS) encounter

uncertainties and noise in tropical regions (Viovy et al., 1992; Atkinson et al., 2012). The GLASS-LAI algorithm provides robust LAI data through tailoring MODIS and CYCLOPES products with general regression neural networks (Liang et al., 2014). In SWAT-T, LAI is simulated for each HRU. The raster cell size of the GLASS-LAI data does not align with the HRUs. We compute the mean GLASS-LAI value for each subbasin by averaging the values of the raster cells that cover it. We then aggregate the simulated LAI at the subbasin scale to match the GLASS-LAI reference. For each subbasin, the subbasin-averaged

LAI value is calculated based on the corresponding HRUs, with the LAI weighted by the area of each HRU within the subbasin. In the model evaluation, we compare the weighted-average subbasin values of the GLASS-LAI to simulated LAI. For AET, we use FLUXCOM-AET (Jung et al., 2019), which combines MODIS, meteorological and eddy covariance tower data to provide a global data set of energy fluxes. Similarly to LAI, the AET data is aggregated at the subbasin scale, with FLUXCOM raster cells weighted by the area of each subbasin. In SWAT-T, AET is simulated at the HRU level and averaged at the

subbasin scale. We compare the weighted-averaged AET data from both FLUXCOM and SWAT-T at the subbasin scale. We use the Kling-Gupta efficiency ($E_{KGE}$) after Gupta et al. (2009), percent bias ($E_{PBIAS}$), and the coefficient of determination ($E_{R2}$) (see Appendix A3 for additional equations) to quantify the SWAT-T performance to model daily discharge, LAI, and AET. Observed streamflow, GLASS-LAI, and FLUXCOM-AET data is respectively available from January 1996 to December 2020, January 2001 to December 2015, and January 2001 to December 2014. Observed precipitation data is available from



**Table 2.** Overview of the data sets that are applied in this study. [1] Observed precipitation data from Rauch et al. (2024) are gridded through nearest-neighbor from the KASS-D database; [2] Observed meteorological data covered solar radiation, temperature, wind speed, and relative humidity.

| Variable | Datasets/Model | Database name or source |
|---|---|---|
| **Data for model set up** | | |
| Digital elevation model | Copernicus GLO-30 (30m x 30m) | Copernicus (2022) |
| Land cover map | Copernicus Global Land Service (100m x 100m) | Buchhorn et al. (2020) |
| Soil map | IMPETUS Soil Data Set | Judex and Thamm (2008) |
| **Model forcing data** | | |
| Precipitation[1] | Gridded (0.05° x 0.05°) from KASS-D | Rauch et al. (2024) |
| Meteorological data[2] | AMMA-Catch network | Galle et al. (2018) |
| ISIMIP3b reference data | W5E5 | Lange et al. (2021) |
| Climate projections | ISIMIP3b: GFDL-ESM4, IPSL-CM6A-LR, MPI-ESM1-2-HR, MRI-ESM2-0, UKESM1-0-LL | Lange (2019) |
| **Data for model performance evaluation** | | |
| Streamflow | AMMA-Catch network (pointwise) | Galle et al. (2018) |
| AET | FLUXCOM (rastered, 0.0833° x 0.0833°) | Jung et al. (2019) |
| LAI | GLASS-LAI (rastered, 250m x 250m) | Liang et al. (2021) |

Januaray 1981 to December 2017. The meteorological data from the AMMA-Catch network is available from January 2006 to December 2020.

We use the daily meteorological data of five GCMs and three scenarios from CMIP6 for the climate impact assessment. The selected scenarios refer to the low, medium-to-high, and high narrative SSP scenarios 1–2.6, 3–7.0, and 5–8.5, respectively, where number before the hyphen represents the shared socio-economic pathway narrative and the number after indicates the

radiative forcing by 2100 in $Wm^{-2}$. We use the climate forcing data from the ISIMIP3b initiative (Lange, 2019) with five GCMs (GFDL-ESM4, IPSL-CM6A-LR, MPI-ESM1-2-HR, MRI-ESM2-0, UKESM1-0-LL), since it provides bias-adjusted, downscaled, and standardized climate data. It has been widely used by the hydrological impact modeling community, enabling cross-sectoral comparisons. For instance, Romanovska et al. (2023) report the applicability of ISIMIP3b data for West Africa, and assess uncertainties and impacts on agriculture for the region.

## 2.4   Sensitivity analysis with the Morris method

Table 1 lists 26 parameters that have been used to assess the vegetation, AET, and discharge simulation with SWAT/SWAT-T. We perform a sensitivity analysis to understand the parameter response complexity on the catchment-scale. Based on a sensitivity ranking, the parameter space can be reduced to the most important parameters. We apply the elementary effects (or Morris) method (Morris, 1991; Campolongo et al., 2007). Generally, the Morris method quantifies the parameter sensitivity





according the elementary effect $d_i$:

$$d_i(q) = \frac{f(q_i,...,q_{i-1},q_i+\Delta,q_{i+1},...,q_k) - f(q)}{\Delta} = \frac{f(q + \Delta e_i) - f(q)}{\Delta}, \tag{3}$$

where $\Delta$ is the parameter step size; $q + \Delta e_i$ is the transformed parameter point; $q = [q_i,...,q_k]$ is any parameter of the total sample size $N$; and $e_i$ consists of a vector of zeros but one in the $i^{th}$ unit vector. For the computation of $d_i$, a total sample size $N$ is generated based on the number of parameters $k$. The parameters $q = [q_i,...,q_k]$ are changed one-at-a-time with

the parameter step size $\Delta$, i.e., one parameter is varied while the others remain constant. The sample size $N = r(k+1)$ is determined based on $r$ defined levels and parameters $q$. The local sensitivity of parameter $q_i$ is quantified as $d_i(q)$. To quantify and rank the sensitivity in this study, the statistical moment $\mu_i$ as mean of the total sample simulation are considered (Morris, 1991). Campolongo et al. (2007) suggest to use the absolute mean $\mu^*$ to not oversee non-monotonic model responses because of opposite signs. The statistical moments for each set $j$ are:

$$\mu_i^*(q) = \frac{1}{r} \sum_{i=1}^{r} |d_i^j(q)|, \tag{4}$$

We apply Latin hypercube sampling to generate a distributed parameter space. We define $r = 500$ and $q = 26$ parameters with which the total sample size is $N = 13500$. The model sensitivity is evaluated with KGE. We quantify the influence of the SWAT-T parameters on streamflow, LAI, and AET separately, denoted as $E_{KGE,Q}$, $E_{KGE,LAI}$, and $E_{KGE,AET}$, respectively. Additionally, we assess the parameter sensitivity on the equally-weighted combination of all three objectives, $E_{KGE,all}$:

$$E_{KGE,all} = 1/3 \cdot E_{KGE,Q} + 1/3 \cdot E_{KGE,LAI} + 1/3 \cdot E_{KGE,AET}. \tag{5}$$

The combination of separated and composed sensitivity analysis objective enhances the understanding of the parameter response complexity. For the application of the Morris method, we integrated the approaches of Morris (1991), Campolongo et al. (2007), and Gupta et al. (2009) into a set of MATLAB scripts.

## 2.5 Three calibration strategies with ROPE, evaluation and robustness check

Table 3 summarizes the calibration strategies with their level of complexity. The approaches consider i) only streamflow at the catchment outlet ("Q only"); ii) streamflow at the outlet and LAI at the subcatchment scale ("Q + LAI"); and iii) streamflow at the outlet, along with LAI and AET at the subcatchment scale ("Q + LAI + AET"). For each calibration strategy, ROPE is applied and a separate objective function $E_{KGE,eff}$ is defined with an equally weighted contribution $E_{KGE,v}$ of the considered variable $v$ (Tab. 3). We integrate the modeling of LAI as proxy of hydrological process representation spatially distributed in

the entire catchment. AET data for model performance evaluation is often limited or the necessary data to compute AET, e.g., with Penman-Monteith, is lacking. Given the LAI–AET linkage in this region (Mamadou et al., 2016), we test whether a calibration approach with streamflow and LAI can be used to predict AET.

The robust parameter estimation (ROPE) algorithm after Bárdossy and Singh (2008) is used for the optimization of the three modeling approaches. ROPE is a depth function-based algorithm to iteratively find parameter sets that give good model





**Table 3.** Overview of model optimization approaches and forcing data.

| Label | Target variables | Forcing data | Objective function |
|-------|------------------|--------------|--------------------|
| Q-o | Q | Observed data | $E_{KGE,eff} = E_{KGE,Q}$ |
| QL-o | Q, LAI | Observed data | $E_{KGE,eff} = 1/2 \cdot E_{KGE,Q} + 1/2 \cdot E_{KGE,LAI}$ |
| QLA-o | Q, LAI, AET | Observed data | $E_{KGE,eff} = 1/3 \cdot E_{KGE,Q} + 1/3 \cdot E_{KGE,LAI} + 1/3 \cdot E_{KGE,AET}$ |
| Q-w | Q | W5E5 data | $E_{KGE,eff} = E_{KGE,Q}$ |
| QL-w | Q, LAI | W5E5 data | $E_{KGE,eff} = 1/2 \cdot E_{KGE,Q} + 1/2 \cdot E_{KGE,LAI}$ |
| QLA-w | Q, LAI, AET | W5E5 data | $E_{KGE,eff} = 1/3 \cdot E_{KGE,Q} + 1/3 \cdot E_{KGE,LAI} + 1/3 \cdot E_{KGE,AET}$ |

predictions and are robust. The ROPE algorithm does not necessarily give the best parameter sets (global optimum), but rather an ensemble of different near-optimum parameters sets (Bárdossy and Singh, 2008). We apply ROPE to derive an ensemble of good parameter sets for climate impact assessment. In addition, the limitations and dependency on a single parameter set (equifinality) are overcome by incorporating multiple well-performing sets. We define "well-performing" if the parameter set gives satisfactory KGE values ($E_{KGE} \geq 0.6$) according to Thiemig et al. (2013).

With ROPE, parameter robustness is quantified using the halfspace depth function introduced by Tukey (1975). Generally, data depth is a quantitative approach to define how central a sampled point is in a multivariate distribution. The more central and, thus, deep a point is located, the less the corresponding objective value differs if the point is subject to a small change, e.g., parameter changes (Bárdossy and Singh, 2008). The calculated parameter set depth value is denoted as $L$ in the following.

To apply the ROPE algorithm, random samples $X_N$ of $N$ parameters and their bounds $d$ are defined. The model is run for
each sample and its performance is quantified. A subset $X_N^*$ of the best performing parameters is defined, e.g., the best 10 % of $X_N$. The performance is evaluated with an objective function (here $E_{KGE,eff}$, Tab. 3). Based on $X_N^*$, $M$ new random samples (labeled $Y_M$) are generated according to the corresponding depth values $L$. A new parameter vector of $Y_M$ is only considered if it lays within the convex hull given by the depth values. The boundary of the convex hull is defined as $L \geq 1$. Each new parameter vector inside the hull ($L \geq 1$) is considered for the sampling of the next iteration. The iterations of the
ROPE algorithm stop as soon as a convergence criteria is reached. We figured that 12 (20) iterations for the "Q only" and "Q + LAI" ("Q + LAI + AET") are needed to guarantee convergence. From one iteration to the next, the 10 % best parameter values are used. After the final iteration, we consider the 20 best parameter sets for the climate impact assessment.

We calibrate against streamflow, GLASS-LAI, and FLUXCOM-AET. The calibration (validation) is defined from January 2008 to December 2011 (January 2012 to December 2015). We use eight years to enable a computational cost-efficient cal-
ibration for the total of six applied model optimizations. The period from 2006 to December 2015 is an adequate overlap of W5E5 and observed data to enable a fair and comprehensive comparison of the forcing data.

The model performance to predict streamflow, LAI, and AET is separately evaluated with $E_{KGE}$ and for each forcing data set. For model validation, we evaluate the efficiencies $E_{KGE}$, percent bias $E_{PBIAS}$, and the coefficient of determination $E_{R2}$ of streamflow, AET, and LAI (see Appendix A3 for supplementary equations).




We assess the reliability of the derived parameter sets through a benchmark evaluation. For hydrological models, benchmarks facilitate an improved interpretation of the model performance and KGE metrics. (Schaefli and Gupta, 2007; Seibert et al., 2018; Knoben et al., 2019, 2020). In this study, we define a lower level of expected model performance based on the interannual mean for every calendar day ("seasonal benchmark") as suggested by Knoben et al. (2020). For this purpose, we compute the daily mean seasonal values for discharge from the observed data of the period 2001 to 2015. For LAI and AET, the mean seasonal values are based on the 8-daily temporal resolution for according to the GLASS-LAI and FLUXCOM-AET data source. Each seasonality is duplicated to be as long as the evaluation period (2008 to 2015), and, thus, represents the identical temporal resolution of the model simulation. We compare the duplicated seasonality to the observed time series and compute $E_{KGE,Q}$, $E_{KGE,LAI}$, and $E_{KGE,AET}$ for the entire evaluation period. We use the duplicated seasonality ("seasonal benchmark") as a synthetic predictor to benchmark SWAT-T's modeling performance.

## 2.6 Model cross validation

The applicability of the parameter sets is evaluated with a cross validation. The parameter sets derived with observed data input are run with model forcing of ISIMIP3b reference data W5E5; and vice-versa the W5E5-parameters with observed data forcing. The cross validation is conducted for the evaluation period (January 2008 to December 2015) and assessed through $E_{KGE,Q}$, $E_{KGE,LAI}$, and $E_{KGE,AET}$. We focus the cross validation on the most comprehensive calibration approach "Q + LAI + AET".

## 2.7 Climate impact assessment

The ISIMIP3b project provides meteorological forcing data from five climate models for three future scenarios, and the corresponding baseline, or historical data. The baseline data set represents the observed climate characteristics, usually for the period from 1901 to 2015, and is used as a reference point to evaluate the future climate scenarios. For meteorological changes, we calculate the mean annual precipitation for each baseline data of the climate models. For each climate model and scenario, we compute the annual precipitation for each year of the near (2031 to 2050) and far (2070 to 2099) future periods. For each period, the mean annual precipitation is calculated. The resulting mean annual precipitation per period is compared to the mean annual precipitation of the baseline data, and the increase or decrease (compared to the baseline data) is determined. For the temperature change, we compute the mean annual temperature for each climate model/scenario and its baseline data. The multi-model mean is calculated from all five GCMs. For hydrological changes, we evaluate the target variables AET and discharge. As baseline for AET (discharge), we use the simulated mean annual AET (discharge) from the period January 2001 to December 2015 from the W5E5 forcing.





# 3 Results

## 3.1 Sensitivity analysis

The sensitivity of the parameters is quantified with the elementary effects method after Morris (1991) whereby a ranking is possible. The higher the $\mu^*$ value of a parameter is, the higher is the parameter sensitivity (Campolongo et al., 2007; Garcia Sanchez et al., 2014). Basically, the elementary effects are calculated according to the model responses, represented as objective function KGE, to the parameter changes. Four different objectives $E_{KGE,Q}$, $E_{KGE,LAI}$, $E_{KGE,AET}$, and $E_{KGE,all}$ are defined to evaluate the parameter sensitivity when modeling discharge, LAI, AET, and the combined performance, respectively, 315 with SWAT-T. Figure 3 shows the values $\mu^*$. The sensitivity analysis considers the entire evaluation period from 2001 to 2015.

**Figure 3.** Ranking of the parameter sensitivity quantified with the elementary effects method. The panels (a), (b), (c), and (d) show the parameter sensitivity with respect to all variables ($E_{KGE,all}$), LAI ($E_{KGE,LAI}$), AET ($E_{KGE,AET}$), and discharge ($E_{KGE,Q}$), respectively. Each panel shows the statistical moment $\mu^*$ for each parameter. We use different marker colors to assign groups for which each parameter is commonly associated with. Please note the different y-axis scaling.





The parameter sensitivity for the equally-weighted objective $E_{KGE,all}$ is displayed in Figure 3a. The ranking of the first 8 parameters is similar to the discharge-only analysis in Figure 3c indicating a strong influence of the discharge performance $E_{KGE,Q}$ in $E_{KGE,all}$. The ranking 9 to 15 is assigned to relevant plant, soil, and AET parameters that are sensitive for $E_{KGE,LAI}$ and $E_{KGE,AET}$.

Figure 3b summarizes the sensitivity ranking with respect to LAI. The LAI parameters (DLAI, T_BASE, FRGRW2, PHU, LAIMX2, LAIMX1; green circles) are generally the most sensitive, together with the SOL_AWC (soil; in purple) and GSI (AET; in red) parameters. The sensitivity ranking confirms the linkage of LAI and AET in the region. The plant growth depends on water and temperature stresses in SWAT/SWAT-T. The water stress is a function of AET and water availability, in which the parameters SOL_AWC and GSI are pivotal. The other soil and AET parameters are less influential, while groundwater and
runoff parameters are irrelevant for predicting LAI.

The sensitivity for the discharge modeling is summarized in Figure 3c. It can be observed that the most sensitive parameters are the groundwater (GWQMN, RCRG_DP, GW_REVAP, REVAPMN) and soil parameters (SOL_AWC, SOL_BD), where GWQMN is most influential. It governs the water depth for which baseflow can occur and partitions the flow into surface and baseflow contributions. Large GWQMN values allow more surface runoff and less baseflow – and vice versa for small values.
The streamflow at the Bétérou gauge is seasonal and connected to the dry and wet precipitation periods. The groundwater parameters, and GWQMN in particular, are essential to model the low flows in the dry season. Figure 3c shows that the groundwater parameters have more impact on the discharge modeling than the runoff (CN_2, CH_K2, HRU_SLP, SURLAG, CH_N2) or plant parameters. Similar findings of the strong groundwater relevance in tropical regions are also reported in the SWAT-T application of López-Ramírez et al. (2021).

The sensitivity ranking in Figure 3d highlights the soil (SOL_AWC, SOL_BD) and AET (ESCO, GSI, EPCO, CAN_MX) parameters to be most sensitive for AET modeling. The AET parameters directly govern the calculation of AET. The soil parameters determine the soil water and availability and, thus, indirectly contribute to the actual amount of AET. The plant parameters are comparably less influential, yet are relevant. The contribution of the plant parameters affirm the significance of LAI for AET in this region. Groundwater and runoff parameters are not substantial for AET prediction.

The sensitivity analysis is applied to identify relevant parameters and reduce the parameter space for a more efficient optimization. Overall, 15 parameters are selected that are substantial for the prediction of discharge, LAI, and AET. The final parameters are listed in Table 1.

## 3.2   Performance of model optimization approaches and robustness check

The models are forced with observed and W5E5 data. The derived parameter sets from the W5E5 forcing are used for the cli-
mate impact assessment. The purposes of the optimization with observed input data are to demonstrate the general performance potential of the ROPE algorithm and to enable a parameter cross validation for the models with W5E5 parameters. We select the 20 best runs from the ROPE optimization for the subsequent model applications. In the following analysis, Figure 4 shows the comparison of simulated and observed discharge, LAI, and AET. Table 4 summarizes the quantitative model performance as KGE values for the calibration and validation periods.





**Figure 4.** Mean annual seasonality of simulated (a) discharge, (b) LAI, and (c) AET from each calibration strategy. Each curve in (a), (b), and (c) represents the mean variables value (discharge, LAI, AET) from n = 20 parameter sets from ROPE. Panels (d), (e), and (f) show scatter plots for discharge, LAI, AET, respectively, of model forcing with observed data. Marker colors indicate the calibration strategy. The mean value of the 20 parameter sets from ROPE is displayed. Label letters represent the calibration variables (Q = discharge, L = LAI, A = AET). The suffix "-o" ("-w") indicates a model forcing with observed data (W5E5 data). Similarly to LAI, panels (g), (h), and (i) show the scatter plots for simulated discharge, LAI, AET, respectively, with model forcing with W5E5 data. The dashed gray 1:1 line in (d) to (i) represents the theoretical best fit. Scatter plots for discharge show daily, for AET and LAI 8-daily points. The time series are presented for the entire evaluation period (2008 to 2015) in all panels.

Overall, the mean annual patterns of discharge, LAI, and AET are moderately well represented in all optimization approaches and with both forcing datasets (Fig. 4a to Fig. 4c). For discharge, the performance of the daily simulation is tied to the forcing data, as spotlighted by the scatter plots in Figure 4d and Figure 4g. The discharge simulation with observed data align well with



the 1:1 line in Figure 4d, i.e., it mimics the recorded streamflow accurately. In contrary, the discharge simulation with W5E5 shows a spread in the scatter, where single high flow events are recorded, but not modeled; and vice-versa for single simulated high flow events. The performance quantification according to KGE is given in Table 4. Model optimization with observed data ($E_{KGE,Q}$ = 0.49 to 0.92, Tab. 4, calibration and validation) outperforms the W5E5 model optimization ($E_{KGE,Q}$ = 0.10 to 0.64, calibration and validation). The variation in performance can be explained with differences in the input precipitation data sets. For instance, the precipitation intensity in 2010 is too low in W5E5 compared to the observation data (see Figure A7). In this period, the simulated discharge is underestimated with W5E5.

**Table 4.** Model performance for daily simulation as KGE [-] for forcing with observation network data and W5E5 data. KGE values are given for the calibration (January 2008 to December 2011) and validation (January 2012 to December 2015) for each strategy. The table lists the minimal, median, and maximal KGE values per variable and calibration (from 20 parameter sets).

| Strategy | Calibration | | | Validation | | |
|---|---|---|---|---|---|---|
| | Q | LAI | AET | Q | LAI | AET |
| **Obs. data forcing** | | | | | | |
| Q | 0.87, 0.88, 0.91 | 0.36, 0.74, 0.85 | 0.53, 0.73, 0.77 | 0.49, 0.81, 0.92 | 0.24, 0.73, 0.81 | 0.59, 0.84, 0.91 |
| Q + LAI | 0.82, 0.84, 0.85 | 0.81, 0.84, 0.88 | 0.71, 0.75, 0.80 | 0.57, 0.74, 0.88 | 0.75, 0.80, 0.82 | 0.81, 0.86, 0.90 |
| Q + LAI + AET | 0.82, 0.85, 0.90 | 0.78, 0.85, 0.87 | 0.75, 0.78, 0.80 | 0.39, 0.85, 0.90 | 0.75, 0.80, 0.83 | 0.86, 0.88, 0.90 |
| **W5E5 forcing** | | | | | | |
| Q | 0.60, 0.62, 0.64 | 0.25, 0.62, 0.86 | 0.50, 0.67, 0.77 | 0.10, 0.43, 0.50 | 0.36, 0.61, 0.81 | 0.55, 0.71, 0.81 |
| Q + LAI | 0.59, 0.61, 0.64 | 0.82, 0.86, 0.88 | 0.68, 0.73, 0.78 | 0.33, 0.45, 0.49 | 0.74, 0.80, 0.82 | 0.72, 0.78, 0.83 |
| Q + LAI + AET | 0.56, 0.60, 0.63 | 0.84, 0.86, 0.88 | 0.71, 0.75, 0.77 | 0.41, 0.46, 0.49 | 0.77, 0.80, 0.83 | 0.77, 0.79, 0.83 |

Figure 4b shows the mean annual LAI pattern from all 20 parameter sets for each optimization approach. Simulated LAI is compared to GLASS-LAI (8-daily) in Figure 4e and 4h for model forcing with observed and W5E5 data, respectively. The performance to model LAI is particularly good in the modeling approaches with LAI as target variable ("Q + LAI" and "Q + LAI + AET") for both forcing data sets. In the growing phase, a small underestimation compared to GLASS-LAI is generally present. However, the plant maturity and leaf senescence are modeled well with "Q + LAI" and "Q + LAI + AET". The performance metrics $E_{KGE,LAI}$ ranges from 0.75 (validation) to 0.88 (calibration) for these approaches (Tab. 4). For "Q only", the LAI modeling is less adequate, particularly for the W5E5 forcing, where the models fail to simulate plant maturity and leaf senescence. The LAI calibration varies from $E_{KGE,LAI}$ = 0.25 to $E_{KGE,LAI}$ = 0.86 (Tab. 4) for discharge-only approaches ("Q-o" and "Q-w").

Similar to LAI, Figure 4c shows the modeled mean annual AET pattern as an average of the 20 parameter sets. Overall, the simulated mean AET mimics well the annual pattern of FLUXCOM reference. The SWAT-T model can represent the two "AET peaks" in the beginning and end of the rainy season, as well as the "AET dip" in the middle of the rainy season. However, the FLUXCOM-AET values are continuously underestimated. It can be observed that "Q only" particularly underestimates the reference AET and even fails to mimic the second "AET peak" due to the lack of LAI representation in this period. The





scatter plots in Figure 4f and Figure 4i confirm SWAT-T's capability to predict AET, but also the simulated underestimation
to FLUXCOM-AET. Generally, the optimization approaches with AET ("Q + LAI + AET") result in very good performance
values for both forcing data sets ($E_{KGE,AET} = 0.75$ to $E_{KGE,AET} = 0.90$, Tab. 4). If the model is calibrated with respect to discharge
and LAI, adequate AET predictions are possible. For the "Q + LAI", $E_{KGE,AET}$ varies between 0.71 to 0.90 (Tab. 4).

The investigation of multiple parameter sets confirms that the modeling performance of SWAT-T is tied to the forcing data,
particularly for discharge. Figure 5 shows the empirical cumulative distribution functions for the 20 best runs from the ROPE
optimization for the calibration period. In the figure, the performance of the seasonal benchmark of discharge, LAI, and AET
is integrated. The performance distribution of the parameter sets to model discharge is evaluated in Figure 5a. Two distinct

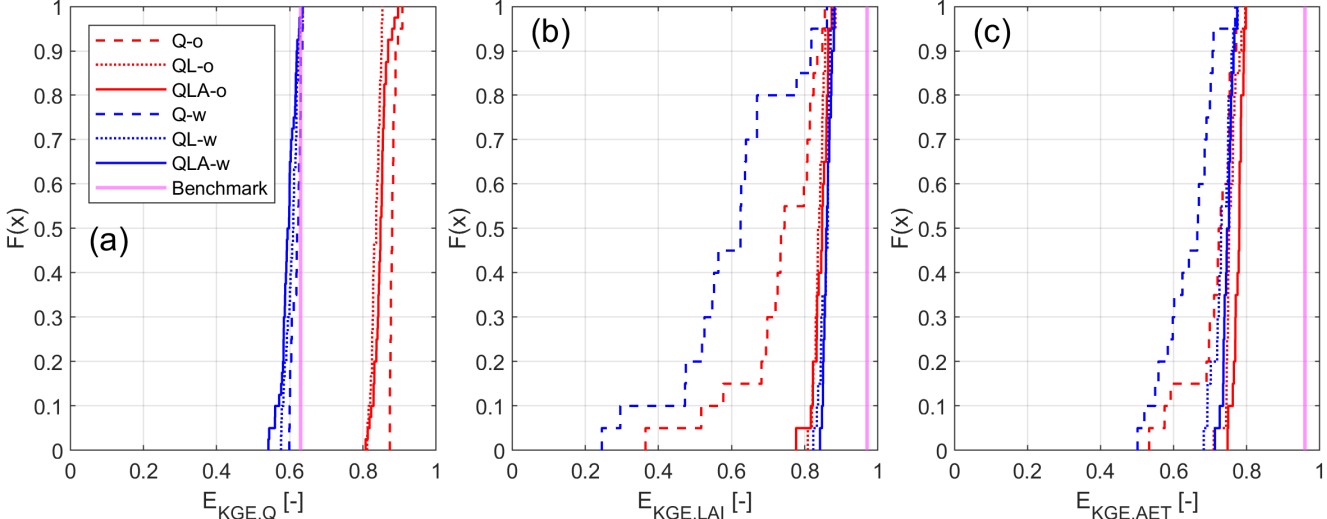

**Figure 5.** Empirical cumulative distribution functions (CDF) for the model performance metrics (a) $E_{KGE,Q}$, (b) $E_{KGE,LAI}$, and (c) $E_{KGE,AET}$.
The CDF in each panel is computed from the KGE values of the 20 best parameter sets per calibration strategy and forcing data. Blue lines
represent the CDFs for W5E5 forcing, red lines for observed data forcing. The line styles indicates the calibration strategy. The label letters
represent calibrations variables (Q = discharge, L = LAI, A = AET). The suffix "-o" ("-w") indicates model forcing with observed data (W5E5
data). The pink line in each panel represents the KGE values for the seasonal benchmark after Knoben et al. (2020).

clusters can be identified: the distribution of the W5E5-based parameters (in blue, with $E_{KGE,Q}$ from 0.54 to 0.64) and based
on the observed data set (in red, with $E_{KGE,Q}$ from 0.81 to 0.91). The W5E5 modeling results in a performance as good as the
seasonal discharge benchmark (pink line, KGE = 0.63). Notably, the model outperforms the seasonal discharge benchmark,
if the observed data set is applied. Figure 5b shows the distribution with respect to $E_{KGE,LAI}$. The values of $E_{KGE,LAI}$ from
discharge-only (Q-o, Q-w) decline rapidly, while the other approaches yield equally good parameter sets for LAI. Figure 5c
displays the distribution functions of $E_{KGE,AET}$. All parameter sets for the "Q + LAI" and "Q + LAI + AET" approaches result
in values of $E_{KGE,AET} \geq 0.68$. In contrary, only a small subset of the "Q-only" approach yields $E_{KGE,AET} \geq 0.68$. Figure 5
highlights that the ROPE algorithm is applicable to derive multiple equally-good parameter sets for LAI and AET modeling.



Yet, the SWAT-T model can't beat the benchmark for LAI or AET. GLASS-LAI and FLUXCOM-AET are reanalysis outputs based on a data assimilation of observations and satellite-based data, such as MODIS data. Both data sets inherit observed seasonal patterns and are processed to mimic reality (Liang et al., 2014; Jung et al., 2019). Their benchmark is thus hard to outperform for a physical-describing model like SWAT-T.

## 3.3    Model cross validation

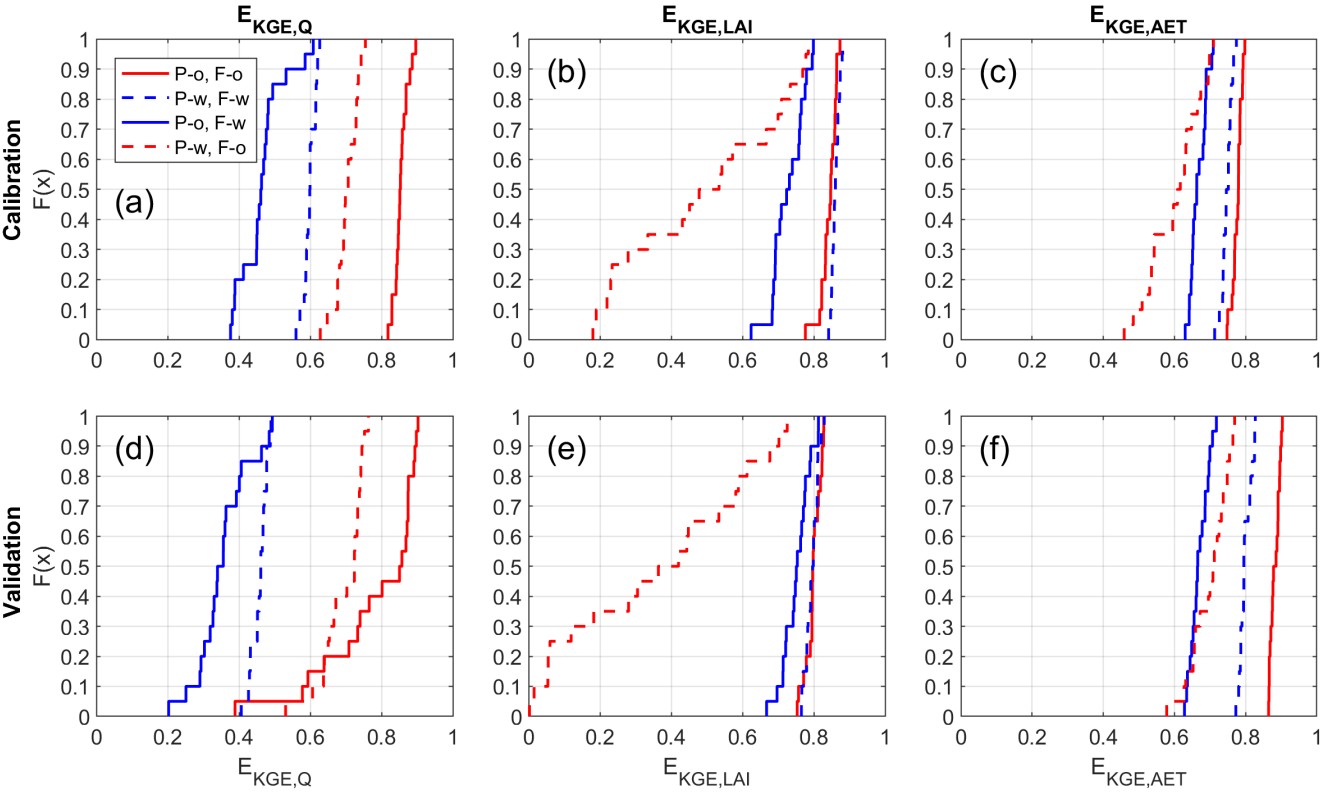

**Figure 6.** Cross validation of the best parameters of the "Q + LAI + AET" calibration strategy. Each line contains n = 20 runs from the optimization with ROPE. The color indicates the model forcing ("F-") data, where red and blue represent forcing with observed ("-o") and W5E5("-w") data, respectively. The line style displays the parameter set origin ("P-"). Straight lines indicate the parameters are derived with observed forcing data ("-o"), dashed lines represent parameter sets from W5E5 forcing ("-w"). The upper and lower row show the performance for the calibration (January 2008 to December 2011) and validation (January 2012 to December 2015) period, respectively. The median of the distribution is equal to F(x) = 0.5.

We assess the applicability of the W5E5-parameters with a parameter cross validation. Figure 6 shows the evaluation of the cross validation for "Q + LAI + AET". If the model is run with W5E5 parameters and observed data forcing (red dashed lines in Fig. 6), the resulting metrics $E_{KGE,Q}$ and $E_{KGE,AET}$ are mainly $\geq 0.6$, which aligns with a satisfactory to good modeling



performance according Thiemig et al. (2013). The median $E_{KGE,Q}$ for the calibration and validation period is 0.70 and 0.69, respectively, which also outperforms the seasonal discharge benchmark (KGE = 0.63). For the prediction of LAI, $E_{KGE,LAI}$

varies for the W5E5-parameters (red dashed lines in Fig. 6b,e), where the median of $E_{KGE,LAI}$ is 0.48 (calibration) and 0.42 (validation). The variation for W5E5 parameters and observed forcing can be explained through the final parameter values (see Tab. A4). For W5E5 optimization, the T_BASE and PHU parameter values are smaller than for observed data forcing. Lower T_BASE and PHU values trigger an earlier start of the plant growth along with a shorter period of maturity. If the input data is replaced, even small temperature changes can influence the growth start and plant maturity.

### 3.4 Future changes of relevant meteorological variables

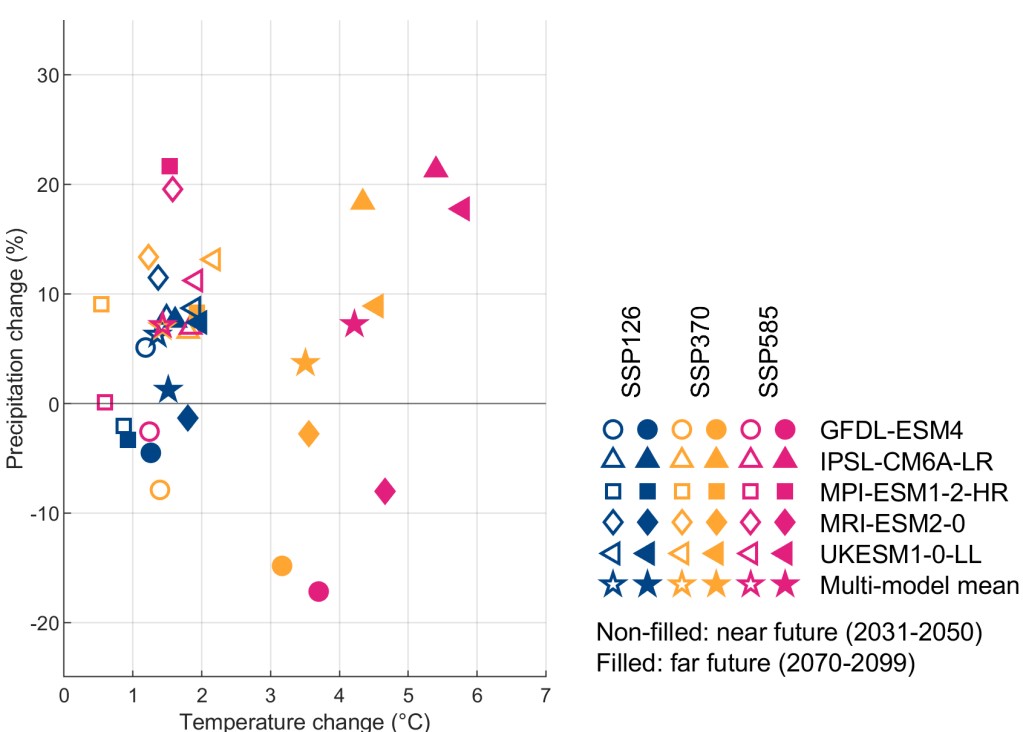

**Figure 7.** Mean projected meteorological changes of the daily mean temperature and annual precipitation for the near (2031-2050; no filled markers) and far (2070-2099, filled markers) future periods compared to the ISIMIP3b baseline (here 1971-2000) for five GCMs and three emission scenarios. The marker colors blue, yellow, and purple refer to SSP1–2.6, SSP3–7.0, and SSP5–8.5, respectively. Multi-model mean is the ensemble mean of the five GCMs.

Figure 7 compares the future changes of precipitation and temperature of the climate models to their baseline data. For all climate models and scenarios, the average temperature is projected to increase between 0.5 to 5.8 °C in the future. All climate models indicate a temperature increase of more than 3 °C for the higher emission scenarios (SSP3–7.0 and SSP5–8.5) in the far future (filled orange and purple symbols), with highest change values of 5.8 °C and 5.4 °C modeled by UKESM1-0-LL and





IPSL-CM6A-LR, respectively. The temperature increase for the low emission scenario (SSP1–2.6) is comparably smaller (0.9 to 2.0 °C) in the near and far future. The multi-model mean temperature change for this region ranges from 1.2 °C (SSP1–2.6, far future) to 4.2 °C (SSP5–8.5, far future). The mean temperature is expected to get warmer especially in the dry period of the year (January to June and October to December), where a prolongation of the dry and hot period (historically from February to May) to January to June is projected (Figure A1).

The precipitation projection is mixed with increases and decreases in the future, where the multi-model precipitation mean varies between increases of 1.2 % (SSP1–2.6) and 7.2 % (SSP5–8.5). For the SSP1–2.6 scenarios, the precipitation changes are ±10 % in the near and far future, except for the near future in the MRI-ESM2-0 model. Positive precipitation changes for selected scenarios are higher than 15 % in UKESM1-0-LL, IPSL-CM6A-LR, MPI-ESM1-2-HR, and MRI-ESM2-0 GCMs. The GFDL-ESM4 model (circles) mainly projects precipitation decreases. The change in precipitation is most substantial in
the rainy season. From all GCMs, an increase in the amount of monthly precipitation is projected for August, September, and October (see Figure A2). The change values are summarized in Tab. A1. The future changes of solar radiation, relative humidity, and wind speed for the near and far future are displayed in Figure A4 to Figure A6.

### 3.5 Climate impact assessment and comparison of modeling approaches

The focus of this comparative study is the analysis of the implications of different modeling approaches on the prediction of
future hydrological changes, particularly of AET and discharge. We evaluate the future discharge and AET response to climatic changes for the near (2031-2050) and far (2070-2099) future period. Generally, the climate change signal for AET is similar across all calibration strategies, where the multi-model mean values of AET indicate future increases. Yet, the future annual AET can vary dependent on the calibration strategy, the underlying GCM, the scenario, and the future period. Figure 8 shows the annual AET for the applied GCM-SSP sets and the multi-model mean of all GCMs for the near and far future period. The
analysis proves that the discharge only strategy ("Q only", blue boxcharts) predicts lower annual AET rates compared to the integrated optimization approaches for each applied GCM-SSP set and for the multi-model mean values for annual AET. The lower rates of AET for "Q only" is furthermore evident if the future AET values are compared to the respective AET baseline. The change rates of annual AET from "Q only" to "Q + LAI + AET" are close for the near future period. While "Q only" predicts an increase of annual AET of 5.5 %, 5.7 %, and 4.2 %, "Q + LAI + AET" results in increases of 7.4 %, 7.2 %, and
5.8 % for the SSP1–2.6, SSP 3–7.0, and SSP5–8.5 scenarios (near period, multi-model mean values, Tab. 5), respectively. For the far future period, the change rates are generally higher for "Q + LAI + AET" than "Q only". However, for the SSP5–8.5 scenario, the future change of AET for "Q only" is 9.2 % compared to an increase of 8.4 % for "Q + LAI + AET". Figure 8 moreover indicates that "Q + LAI" results in predictions of annual AET which are close to "Q + LAI + AET". Thus, "Q + LAI" can approximate AET rates on the catchment scale even if AET is not integrated in the parameter estimation. The LAI
computation can be used as a proxy for AET because the plant growth and AET processes are directly connected in SWAT-T.

The future annual AET rates also depend on the GCM model. For the near future (Fig. 8), the range of predicted annual AET is smaller (750 to 1000 mm) than compared to the far future GCM model applications (710 to 1120 mm, Fig. 8). The modeling with GFDL-ESM4 and MPI-ESM-1-2-HR result in comparably smaller annual AET rates, while the simulations



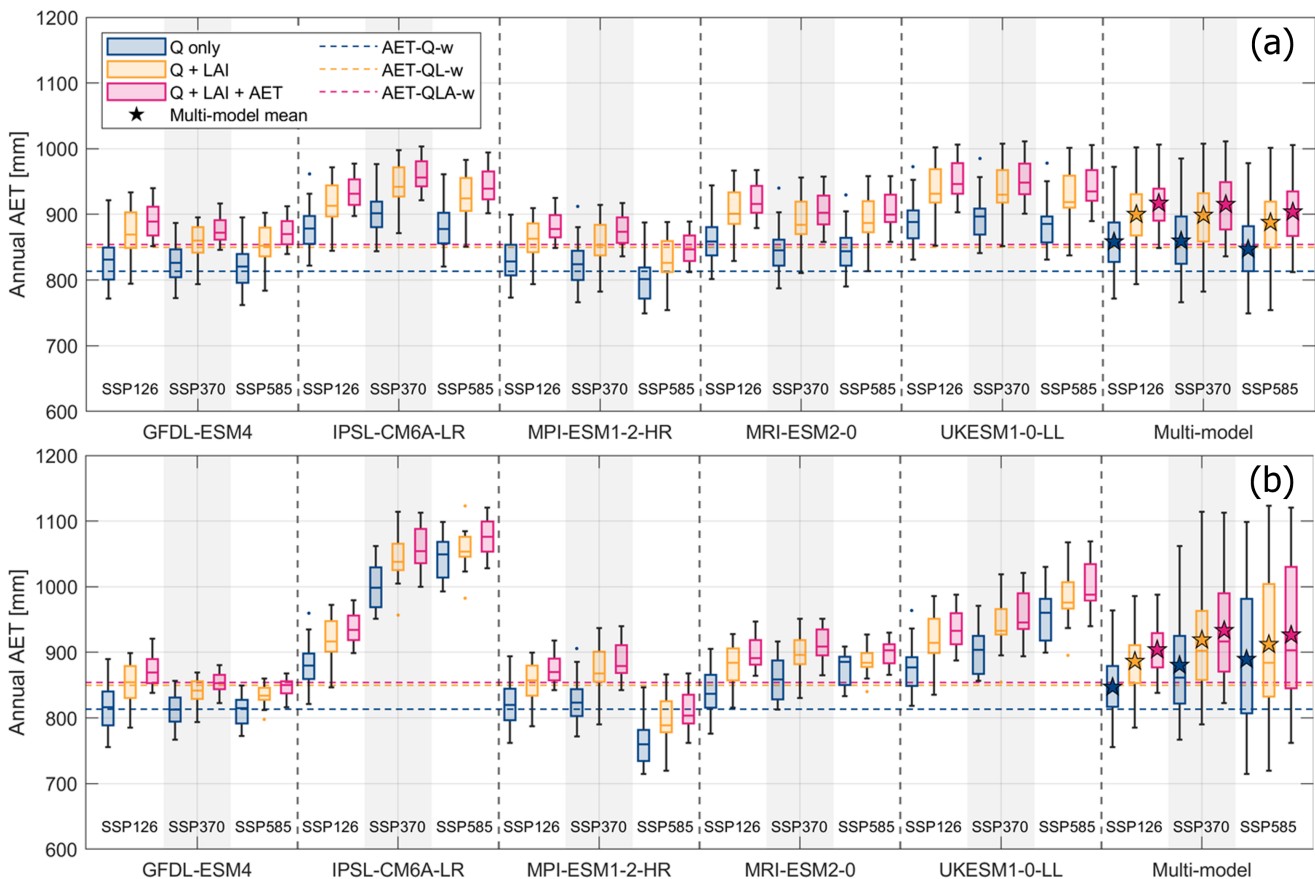

**Figure 8.** Future annual AET simulated with the five GCM and three SSP scenarios and the multi-model ensemble evaluation for (a) the near-future period (2031-2050) and (b) the far-future period (2070-2099). Colors indicate the calibration "Q only"/"Q-w" (blue), "Q + LAI"/"QL-w" (yellow), and "Q + LAI + AET"/"QLA-w" (pink). Each box for GCM-SSP contains n = 20 projected mean annual AET values from the 20 best parameter sets. For the multi-model ensembles, each box contains n = 100 values (5 GCMs x 20 parameter sets). Stars highlight the multi-model mean values for each calibration process and SSP scenario. Dashed horizontal lines indicate the simulated mean annual AET from the period 2001 to 2015 with W5E5 forcing. Colors represent the corresponding calibration strategy. Label letters for the horizontal lines indicate the calibrations variables (Q = discharge, L = LAI, A = AET). The suffix "-w" stands for the forcing with W5E5.

forced by IPSL-CM6A-LR, MRI-ESM2-0, and UKESM1-0-LL indicate higher annual AET values for the near future. For the far future period, high annual AET rates are particularly predicted with IPSL-CM6A-LR. The GCM forecasts for the far future an increase in precipitation and temperature (Fig. 7), which translates to available water and energy for AET. The simulation with forcing data from the MPI-ESM2-0 model shows distinct differences between the SSP scenarios, where the also future decreases of annual AET rates can occur. For the far future, the MPI-ESM2-0 model simulates the low temperature changes ($\Delta T \leq 2$ °C, Fig. 7), and, thus, the increase in AET is the lowest with MPI-ESM2-0 compared to the other GCM applications.





**Figure 9.** Future monthly AET values for the 5 GCMs and emission scenarios (a) SSP1–2.6, (b) SSP3–7.0, and (c) SSP5–8.5 for the near-future period (2031-2050). Each box contains n = 100 monthly AET values (5 GCMs x 20 parameter sets). Colors indicate the calibration process "Q only" (blue), "Q + LAI" (yellow), and "Q + LAI + AET" (pink). The crosses represent the monthly mean AET values from the corresponding calibration strategy.

The differences in predicting annual AET are explained with variations in monthly AET dynamics. Figure 9 and Figure 10 show the monthly mean AET for the calibration approaches for the near and far future period, respectively. The figures also display the monthly mean AET values from the each model optimization forced with W5E5 data (black cross). The analysis shows for which months the calibration approaches predict different AET rates. For the near future period (Fig. 9), the monthly AET rates predicted with "Q only" are lower in the period from August to December. This period describes the end of the wet

and transition into the dry period. Given the discharge-centered optimization, the model does not represent the plant and soil water processes in this period. The smaller AET rates for "Q only" are evident for all three SSP scenarios in the near future period. Concurrently, the predicted mean seasonal values from "Q + LAI" mimics the "Q + LAI + AET" where only small differences per month are computed.





**Figure 10.** Future monthly AET values for the 5 GCMs and emission scenarios (a) SSP1–2.6, (b) SSP3–7.0, and (c) SSP5–8.5 for the far-future period (2070-2099). Each box contains n = 100 monthly AET values (5 GCMs x 20 parameter sets). Colors indicate the calibration process "Q only" (blue), "Q + LAI" (yellow), and "Q + LAI + AET" (pink). The crosses represent the monthly mean AET values from the corresponding calibration strategy.

For the far future period (Fig. 10), the pattern of monthly mean AET differences is similar to the predictions of the near future period. The monthly AET rates of "Q only" are underestimated from August to December. The analysis for the far future period moreover shows large variations within each calibration strategy itself. The box charts are wider in the far period, than in the near period evaluations, particularly for SSP5-8.5 (see Fig. 10c). The chart sizes indicate that within each calibration strategy, the range of monthly AET varies, and, thus, the forcing from the GCMs is varying, too. The comparison of future monthly mean AET values to the monthly AET rates from the W5E5 optimization points out differences particularly in April for all calibration approaches and SSP scenarios. The contrasting high rates of future monthly AET in April is related to the strong increase of temperature in the future spring months (see Fig. A1). In April, the warmer climate overlaps with the onset of the rainy season, which eventually catalyzes the AET generation.




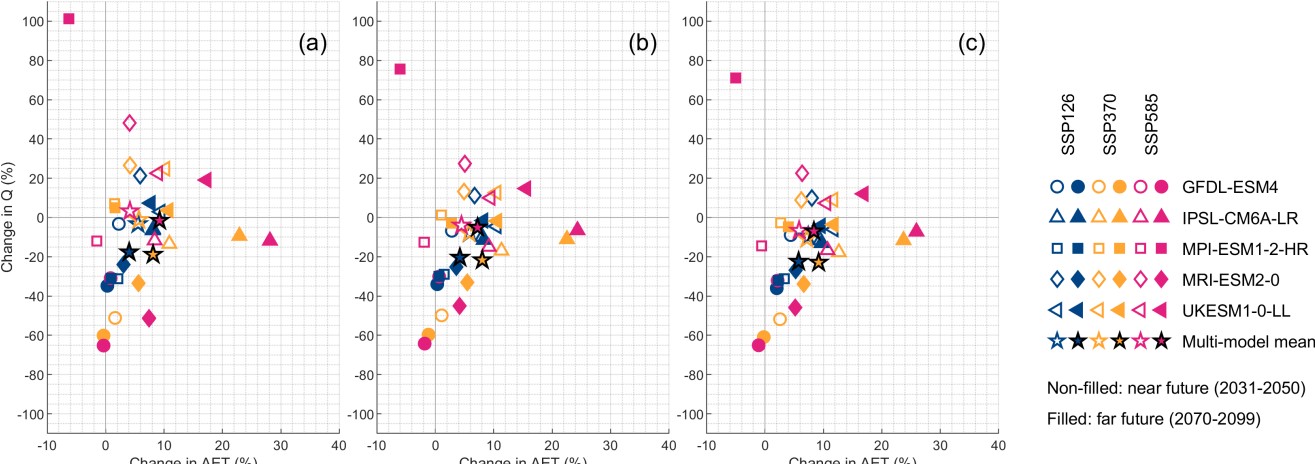

**Figure 11.** Mean future changes in annual AET and discharge for the GCM and emission scenarios as well as the multi-model mean for the calibration approaches (a) "Q only", (b) "Q + LAI", and (c) "Q + LAI + AET". Each marker represents the mean change computed from the 20 best parameter runs. The colors indicate the emission scenario. The markers represent the GCM. Filled (non-filled) corresponds to the far-future (near-future) period.

Generally, the future rate of AET is computed to increase for the Bétérou Catchment based on the GCMs integrated in the ISIMIP3 project. Depending on the calibration strategy and the SSP scenario, the multi-model mean changes range from 4.0 %
("Q only", SSP1–2.6, far period) to 9.2 % ("Q + LAI + AET", SSP3–7.0, far period; and "Q only", SSP5–8.5, far period) as indicated from the multi-model mean values in Figure 11 and Tab. 5. The projected changes of AET (multi-model mean) are lower for "Q only" than "Q + LAI + AET". The overview of future changes in Figure 11 and Tab. 5 indicates high AET increases particularly for the IPSL-CM6A-LR, which projects high precipitation and temperature increase. Negative AET changes are computed only partially and mainly with forcing from MPI-ESM-1-2-HR (Figure 11a).

For discharge, the future changes can be positive and negative depending on the underlying GCM forcing and calibration strategy. The application of the MPI-ESM1-2-HR model data results in a strong increase of discharge (up to 101.1 %, SSP5–8.5, "Q only"). The forcing data of GFDL-ESM4 yields mostly decreases of future discharge due to projected precipitation decreases and temperature increases of >3 °C. For the near future, the discharge changes from "Q only" range from −3.3 to 3.2 % (multi-model mean, Tab. 5). For "Q + LAI" and "Q + LAI + AET", the multi-model mean changes of discharges are
simulated from −7.9 to −4.0 % and from −10.8 to −6.7 %, respectively. For the far-future period, the multi-model mean changes of discharge show a substantial decrease independent of the calibration strategy, where the change rates for "Q + LAI + AET" are the lowest due to the higher simulated evaporative demand. The changes are −18.9 to −1.6 % for "Q only", −21.7 to −5.1 % for "Q + LAI", and −22.9 to −7.0 % for "Q + LAI + AET".



**Table 5.** Mean future changes in annual AET $\Delta E$ and discharge $\Delta Q$ for the GCM and emission scenarios as well as the multi-model mean for the calibration approaches "Q only", "Q + LAI", and "Q + LAI + AET". The mean changes are computed from the 20 best parameter runs. The annual AET changes are denoted as $\Delta E$ in this table for readability.

| GCM | Q only | | | | Q + LAI | | | | Q + LAI + AET | | | |
|---|---|---|---|---|---|---|---|---|---|---|---|---|
| | Near | | Far | | Near | | Far | | Near | | Far | |
| | $\Delta Q$ | $\Delta E$ | $\Delta Q$ | $\Delta E$ | $\Delta Q$ | $\Delta E$ | $\Delta Q$ | $\Delta E$ | $\Delta Q$ | $\Delta E$ | $\Delta Q$ | $\Delta E$ |
| **SSP1–2.6** | | | | | | | | | | | | |
| GFDL-ESM4 | −3.2 | 2.3 | −34.8 | 0.3 | −6.8 | 2.8 | −34.0 | 0.3 | −9.0 | 4.4 | −36.0 | 2.0 |
| IPSL-CM6A-LR | −6.5 | 8.1 | −6.0 | 8.1 | −9.8 | 8.0 | −11.4 | 8.3 | −11.6 | 9.5 | −13.1 | 9.7 |
| MPI-ESM-1-2-HR | −31.1 | 2.1 | −31.1 | 0.9 | −29.0 | 1.5 | −29.9 | 0.6 | −31.2 | 3.3 | −32.0 | 2.3 |
| MRI-ESM2-0 | 21.4 | 5.9 | −23.9 | 3.1 | 11.1 | 6.7 | −25.2 | 3.6 | 9.8 | 8.1 | −26.9 | 5.2 |
| UKESM1-0-LL | 2.9 | 9.4 | 7.3 | 7.7 | −4.5 | 10.5 | −1.6 | 8.2 | −6.0 | 11.8 | −4.3 | 9.6 |
| Multi-model mean | −3.3 | 5.5 | −17.7 | 4.0 | −7.8 | 5.9 | −20.4 | 4.2 | −9.6 | 7.4 | −22.4 | 5.8 |
| **SSP3–7.0** | | | | | | | | | | | | |
| GFDL-ESM4 | −51.2 | 1.6 | −60.1 | −0.4 | −49.9 | 1.1 | −59.7 | −1.2 | −51.8 | 2.6 | −60.9 | −0.2 |
| IPSL-CM6A-LR | −13.4 | 10.9 | −9.4 | 22.9 | −16.9 | 11.3 | −11.0 | 22.5 | −17.7 | 12.6 | −11.6 | 23.7 |
| MPI-ESM-1-2-HR | 7.0 | 1.6 | 4.9 | 1.6 | 1.3 | 1.0 | −3.0 | 2.8 | −2.6 | 2.7 | −4.7 | 4.0 |
| MRI-ESM2-0 | 26.7 | 4.2 | −33.5 | 5.7 | 13.1 | 4.9 | −33.1 | 5.5 | 8.8 | 6.3 | −33.8 | 6.7 |
| UKESM1-0-LL | 24.8 | 10.2 | 3.7 | 10.8 | 12.7 | 10.5 | −1.8 | 10.6 | 9.1 | 11.7 | −3.6 | 11.8 |
| Multi-model mean | −1.2 | 5.7 | −18.9 | 8.1 | −7.9 | 5.8 | −21.7 | 8.0 | −10.8 | 7.2 | −22.9 | 9.2 |
| **SSP5–8.5** | | | | | | | | | | | | |
| GFDL-ESM4 | −30.9 | 0.9 | −65.2 | −0.3 | −30.1 | 0.7 | −64.2 | −1.8 | −32.1 | 2.2 | −65.1 | −1.1 |
| IPSL-CM6A-LR | −11.7 | 8.4 | −11.8 | 28.1 | −14.9 | 9.2 | −6.5 | 24.3 | −16.6 | 10.7 | −7.2 | 25.9 |
| MPI-ESM-1-2-HR | −12.0 | −1.5 | 101.1 | −6.3 | −12.7 | −2.0 | 75.6 | −6.1 | −14.5 | −0.6 | 71.0 | −5.0 |
| MRI-ESM2-0 | 48.1 | 4.1 | −51.2 | 7.4 | 27.5 | 5.1 | −45.1 | 4.2 | 22.5 | 6.4 | −45.9 | 5.2 |
| UKESM1-0-LL | 22.5 | 9.0 | 19.1 | 17.3 | 10.0 | 9.5 | 14.8 | 15.6 | 7.3 | 10.5 | 12.1 | 16.9 |
| Multi-model mean | 3.2 | 4.2 | −1.6 | 9.2 | −4.0 | 4.5 | −5.1 | 7.2 | −6.7 | 5.8 | −7.0 | 8.4 |

# 4 Discussion

## 4.1 Optimization strategy and hydrological modeling

This study primarily focus on the variability of future AET depending on the calibration strategy in hydrological modeling. For this purpose, three calibration methods with different levels of complexity and information indices ("Q only", "Q + LAI", and "Q + LAI + AET") are evaluated. Similar to previous comparative studies (Huang et al., 2020; Ismail et al., 2020; Koch et al., 2020; Mishra et al., 2020; Wen et al., 2020), the analysis of simple to comprehensive calibration strategies is investigated, in





this study with particularly for AET. The different calibration complexities from "Q only" to "Q + LAI + AET" also serve as a proxy for a benchmark test with lower and upper arguments, as suggested by Seibert et al. (2018). The integration of an "AET only" approach may add another layer of "upper benchmark". Yet, we also assess the calibration strategy impacts on discharge changes for the region. Disregarding discharge in the hydrological modeling can imply large discharge variability (Odusanya et al., 2019; Sarmiento et al., 2025). We have thus avoided to include "AET only" to minimize discharge uncertainties.

Model evaluation in this study is based on in-situ monitoring and satellite-derived reanalysis datasets (GLASS-LAI and FLUXCOM-AET). GLASS-LAI and FLUXCOM-AET have been validated by Merk et al. (2024) for two distinct locations within the Bétérou Catchment. GLASS-LAI and FLUXCOM-AET rely on models and assumptions, and are provided at a coarser resolution than the hydrological model. To address scale differences, the SWAT-T outputs, GLASS-LAI and FLUXCOM-AET data were spatially projected onto the same units. While this approach allows for a spatially explicit eval-
uation that reflects land cover variability within the Bétérou catchment, it also introduces some information loss, e.g., from the aggregation of AET heterogeneity. The subbasin-level evaluation offers a practical compromise, where spatial context is preserved at a scale appropriate for hydrological modeling.

The hydrological cycle in the Western African region is driven by precipitation, AET, and discharge (Rodell et al., 2015). Although soil moisture is a key internal state variable in hydrological modeling, we have not included soil moisture products
for the model validation. Accurate simulations of the dynamics of AET and discharge has been shown to reliably represent the hydrological behavior in the Ouémé catchment (Odusanya et al., 2021). We thus focused the model evaluation on discharge, AET, and LAI, and excluded soil moisture to minimize additional uncertainty and potential conflicts between observational variable constraints.

We integrate LAI as a proxy for vegetation in the hydrological modeling. The relevance of LAI on the AET and discharge
estimation in sub-humid regions has been highlighted prior, e.g., in Alemayehu et al. (2017); Hoyos et al. (2019); Ferreira et al. (2021); López-Ramírez et al. (2021). Previous studies for the region, like Poméon et al. (2018) or Odusanya et al. (2021), include AET in the hydrological modeling of the Ouémé River in Benin, while others have focused on "Q only" approaches (Bossa et al., 2014; Danvi et al., 2018). We demonstrate that a detailed LAI modeling can predict adequate AET outputs.

The evaluation of modeled discharge from the ROPE application yields different KGE ranges depending on the forcing data.
For calibration with data from the observation network, the maximum $E_{KGE,Q}$ (validation) is 0.91 (0.92), 0.85 (0.88), and 0.90 (0.90) for "Q only", "Q + LAI", and "Q + LAI + AET", respectively. These results align with findings from other studies; for instance, Afféwé et al. (2025) report 0.83 (0.77) with HBV model, or Herzog et al. (2021) validate the Parflow-CLM model with a mean KGE value of 0.82. The strong KGE performance confirms that the ROPE algorithm and the SWAT-T are well-suited for discharge calibration in the Bétérou Catchment.

However, it is still essential to calibrate the hydrological impact model with the W5E5 forcing dataset if the bias-adjusted ISIMIP3b GCM outputs are used. In this study, the maximum $E_{KGE,Q}$ from calibration (validation) are 0.64 (0.50), 0.64 (0.49), and 0.63 (0.49) for "Q only", "Q + LAI", and "Q + LAI + AET", respectively. We assume the KGE values are smaller compared to the observed data due to the W5E5 data forcing, particularly precipitation. The W5E5 data set represents the seasonality accurately well, but it lacks the depiction on sub-seasonal variations and single rainfall events for the Bétérou Catchment





in West Africa. Similar results with CMIP6 are reported by Wang et al. (2022) for the Congo River in Africa who figured challenges in discharge representation through CMIP6 climate data. However, we benchmark the discharge performance with the seasonal benchmark after Knoben et al. (2020), which shows an adequate behavioral modeling of discharge. Combined with the parameter cross validation, the W5E5-models are applicable and can still represent the mean seasonal patterns of the hydrological cycle in Bétérou. We thus recommend benchmarking and cross validation with observed data if the ISIMIP3b is

used for future studies in West Africa.

The model optimization and selection of the 20 parameter sets for the climate impact assessment are based on the performance metrics KGE, which is a robust criteria to asses the runoff generated within a catchment (Gupta et al., 2009). Single metrics can however overlook pitfalls of the model structure and defer the model and parameter choice (Schaefli and Gupta, 2007; Knoben et al., 2019; Hallouin et al., 2020; Cinkus et al., 2023). We apply multiple parameter sets, the benchmarking after

Knoben et al. (2020), and parameter cross validation to increase the reliability of the single-metrics based model choice for climate impact assessments. Looking ahead, the model optimization can include further performance criteria to assess specific features of hydrological process in the region, such as the Nash-Sutcliff Efficiency (NSE) or the logarithmic NSE, among many others, to better include high or low flows, respectively.

## 4.2   Climate future data and meteorological changes

We used five GCMs with three SSP scenarios from ISIMIP3b (based on CMIP6) equally weighted, where we followed the conventional model democracy approach. Model democracy in climate impact studies can be crucial (Knutti, 2010). However, the use of multiple GCM-scenario combinations helps to reduce structural GCM model uncertainty, while enhancing transparency and comparability. For example, the application of CMIP6 data in this study for the Bétérou Catchment in Benin enables comparisons with similar assessments in this region, such as the findings of Yonaba et al. (2023) for the Tougou Basin in Burkina

Faso. Both studies indicate substantial increases in projected discharge under the MPI-ESM1-2-HR model, with changes up to 101.1 % in this study and up to 117 % in (Yonaba et al., 2023).

However, assuming all GCM models perform reasonably well globally is critical, since poorly performing GCMs can influence ensemble-based hydrological impact statements (Flato et al., 2014). Hattermann et al. (2018) highlight that GCMs are major source of uncertainty in the climate-hydrological impact modeling. Consequently, the influence of the calibration

strategy on future AET is likely to be smaller compared to the GCM model uncertainty. Nevertheless, our study demonstrates that the future changes of AET strongly depend on the calibration strategy. Thus, disregarding AET during model calibration may introduce an additional layer of uncertainty other than the GCM uncertainty.

The application of multiple ISIMIP3b GCMs offers a transparent and standard way to understand climate changes and hydrological responses across studies, and to avoid uncertainties from individual downscaling and bias corrections. Romanovska

et al. (2023) demonstrate the potential benefits of the ISIMIP3b climate data since its multi-model mean ensembles can even outperform individual models when compared to observed precipitation in Western Africa. The seasonal patterns of temperature and precipitation in the GCM historical data closely align with those observed in the monitoring network. The monthly values are similar, and qualitatively meet the criteria suggested by (Krysanova et al., 2017; Hattermann et al., 2018). The me-





teorological changes for the Bétérou Catchment in Benin indicate a clear projection for temperature, but a divers picture for
precipitation. For temperature, all GCMs forecast increases of 0.5 to 5.8 °C, which generally aligns with the changes projected
with CMIP6 for the West African Region (Almazroui et al., 2020).

The projected precipitation derived from the ISIMIP3b data set indicates a diversified scatter of changes. The multi-model
mean values indicate small (1.2 %, SSP1–2.6) to large (7.2 %, SSP5–8.5) precipitation increases. Yet, the variability between
single models is distinct. Substantially wetter periods are indicated by MPI-ESM1-2-HR, IPSL-CM6A-LR, and UKESM1-
0-LL, while GFDL-ESM4 and MRI-ESM2-0 project drier future rainfall. The ensemble spread in CMIP6 precipitation pro-
jections can be attributed to differences in how models simulate the West African Monsoon (Almazroui et al., 2020). Similar
precipitation variability for West Africa has also been reported in other climate forcing datasets. For example, Nikulin et al.
(2012) found substantial spreads in CORDEX precipitation projections that are also linked to the complex monsoon dynamics.

## 4.3 Hydrological changes

The annual AET is generally predicted to increase independent of the GCM, the time period (near-, far future), or the calibration
strategy ("Q only", "Q + LAI", "Q + LAI + AET"). For the low emission scenario (SSP1–2.6), the multi-model mean changes
range from 4.0 % to 7.4 % across all calibration strategies. The high emission scenario (SSP5–8.5) predicts multi-model mean
changes of 4.2 % to 9.2 %. Previous impact studies for the Ouémé region report increases, but also decreases in future AET for
the region (Bossa et al., 2012, 2014; Danvi et al., 2018). However, these studies relied on models calibrated against discharge,
without explicitly including AET in the calibration. As our results demonstrate, disregarding AET from calibration can defer
the simulated amount of future AET, where "Q only" simulates lower future annual AET changes than "Q + LAI" or "Q + LAI
+ AET". These findings align with Mishra et al. (2020), who demonstrate the influence of the AET calibration strategy on the
climate impact assessment for the Upper Godavari in India.

Differences in projected AET across the calibration approaches are most pronounced between August and December. The
calibration strategy has a smaller impact on AET simulation during the early part of the year, which is the shift from dry to wet
period. The seasonal AET pattern in Bétérou is characterized by two peaks (April to June and September to November) with a
notable dip during the peak rainy season (June to September). This mid-season decline is primarily caused by cloud cover and
rainfall, which limit solar radiation and thus reduce AET. SWAT-T can simulate AET dynamics in the beginning of the season
(January to July), which is largely independent of the calibration strategy. However, from August onward, the AET simulation
becomes more sensitive to the calibration strategy, demonstrating the need for carefully adjusted model parameters. The soil
(e.g., SOL_AWC, SOL_BD), AET (e.g., GSI), and plant growth (DLAI, T_BASE, FRGRW2) parameters can be essential to
realistically represent post-monsoon and dry-season AET in Bétérou.

The analysis of future discharge changes shows increases and decreases depending on the GCM, emission scenario, and eval-
uation period. The different projections are largely attributed to the complexity of how GCMs simulate precipitation dynamics
and projections of the West African Monsoon (Paeth et al., 2011). Consequently, the model forcing with MPI-ESM-1-2-HR
outputs can also result in high future discharges (change of 101.1 % for "Q only", SSP5–8.5, far future). Similar substantial
discharge changes with MPI-ESM-1-2-HR are also reported by Yonaba et al. (2023) for a catchment in Burkina Faso. This





GCM projects substantial precipitation increases with minor temperature changes for the West African region. Overall, the analysis of the multi-model mean for future discharge changes essentially indicates a decrease of discharge for the Bétérou region. The multi-model mean changes vary from 22.9 % to 3.2 %. Discharge decreases for the Ouémé region have also been found in other studies (Bossa et al., 2012, 2014; Danvi et al., 2018). The decreases can potentially be attributed to the increase in temperature, which intensify the evaporative demand.

In this study, the evaluation of hydrological changes is focused on the primary components of the hydrological cycle AET and discharge in the Bétérou Catchment. These variables provide a robust basis for interpretation, since they are reasonably well constrained by observations. Other water balance components, such as the partitioning into surface runoff, lateral flow, and baseflow, are not specifically investigated at this stage. Soil moisture dynamics are not analyzed in detail, because no particular calibration or validation has been conducted. Looking ahead, a comprehensive analysis of the water balance components will be conducted with a special focus on soil moisture, the partitioning of the flow components, and hydrological variables.

## 5 Conclusion

The broad implication of this research is the demonstration of how different the response of future AET to climate change can be depending on the calibration strategy with an ecohydrological impact model. Although being relevant for the hydrological cycle and water availability estimation, AET often remains disregarded in model calibration in the framework of climate impact assessment in West Africa. We investigate the role of different calibration strategies on future AET projections under different GCM and emission scenario combinations for the demonstration site Bétérou in West Africa using the ecohydrological model SWAT-T.

This study couples comprehensive parameter estimation with climate impact assessment, and uses multiple near-optimal parameter sets for each calibration strategy. This ensemble-based approach avoids reliance on a single best-fit solution and explicitly accounts for model parameter equifinality. This way, the robustness of the future projections is increased and the influence of the calibration strategy can be spotlighted.

The analysis shows that the calibration strategy can influence the future AET projections. The simulated mean annual AET substantially varies depending on the calibration strategy for the baseline (2001 to 2015) and future periods (2031 to 2050 and 2070 to 2099). The annual AET is smaller for discharge-only calibration compared to "Q + LAI" and "Q + LAI + AET" approaches. The relative future AET changes of the GCM-ensemble mean for low emission scenarios (SSP1–2.6) range from $\Delta E = 4.0\%$ (discharge only) to $\Delta E = 7.4\%$ (AET in calibration) in the near-future period. For high emission scenarios (SSP5–8.5) and the far-future period, the future change of AET is $\Delta E = 9.2\%$ (disregarding AET in model calibration) to $\Delta E = 8.4\%$ (AET in calibration). The differences in future AET changes from the calibration strategy are tied to the simulation of the mean seasonal AET values. Disregarding AET in calibration results in lower AET predictions for the wet to dry transition period (August to December). This study moreover highlights the value of plant growth (LAI) for AET estimation. Given the high transpiration rates in the region, vegetation dynamics (LAI) can serve as a proxy for AET when direct calibration is not feasible.



This study aligns with the well-document findings the application of the ISIMIP3b (CMIP6) data can have for climate impact assessment in West Africa. The baseline forcing data set W5E5 captures the precipitation pattern of the Bétérou Catchment reasonably well, yet can have implications in representing individual rainfall events. For instance, model calibration using W5E5 forcing shows lower performance compared to simulations driven by observed meteorological data for discharge simulation. Additionally, some GCMs can be more sensitive to the regional affecting West African Monsoon. This leads to higher precip-

itation projections, which consequently results in high future discharge increases (up to $\Delta Q = 101.1\%$ for certain GCMs; "Q only").

    In hydrological climate impact assessment, GCM uncertainty remains most challenging. Yet, our findings underscore the value of a robust hydrological model calibration for impact assessment. We address the model and parameter uncertainties in the climate-hydrological modeling chain, and highlight how disregarding detailed AET calibration can lead to biased AET

impact assessments. The presented methodology can be transferred to other AET-dominant regions. Looking ahead, integrating additional remotely sensed or satellite-based data, such as soil moisture or ensemble AET data, can further reduce modeling uncertainties and improve the reliability of future AET and discharge assessments.





# Appendix A: Appendix

## A1 Appendix A1 - Figures

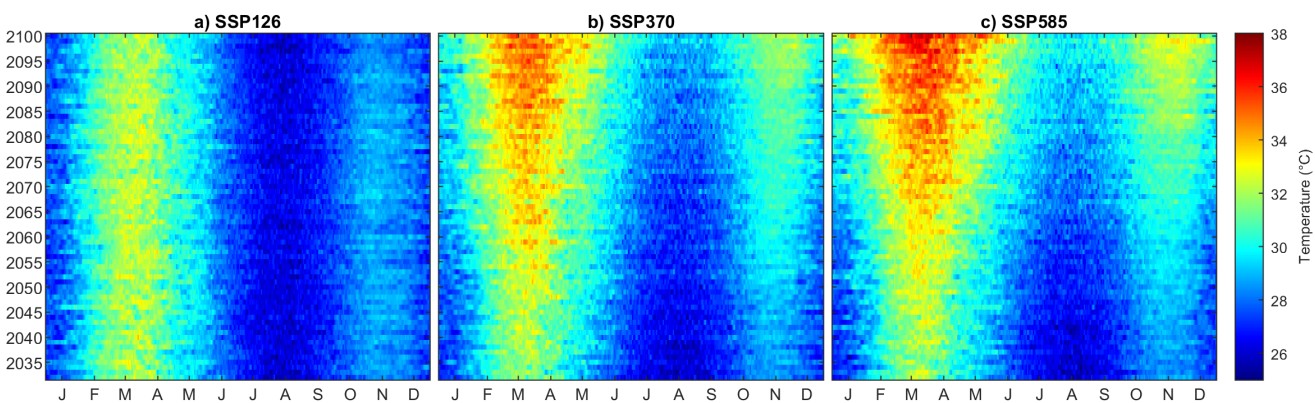

**Figure A1.** Daily average temperature projections (multi-model mean) in three climate change scenarios (SSP1–2.6, SSP3–7.0, SSP5–8.5).

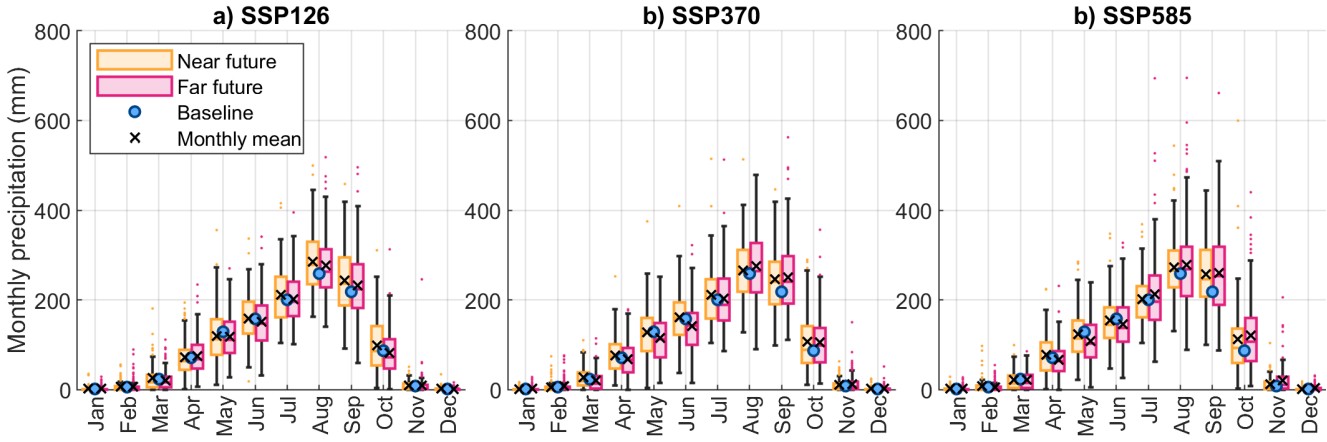

**Figure A2.** Seasonal cycle of monthly precipitation projections (multi-model mean) in three climate change scenarios (SSP1–2.6, SSP3–7.0, SSP5–8.5). Blue circles represent the monthly mean precipitation from ISIMIP3b baseline (1951 to 2015). Black crosses are used to highlight the multi-model monthly mean values.




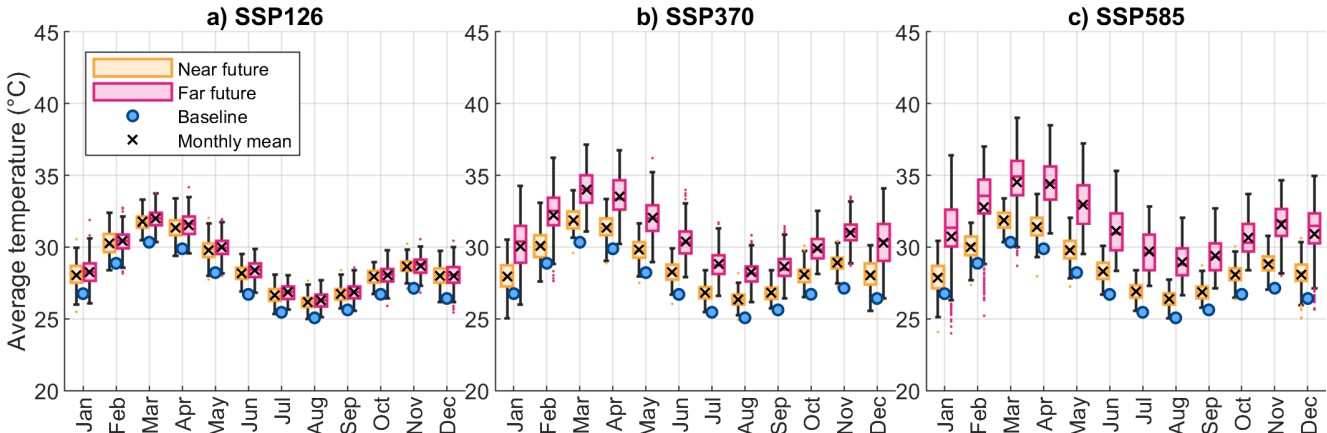

**Figure A3.** Seasonal cycle of average temperature projections (multi-model mean) in three climate change scenarios (SSP1–2.6, SSP3–7.0, SSP5–8.5). Blue circles represent the monthly mean temperature from ISIMIP3b baseline (1951 to 2015). Black crosses are used to highlight the multi-model monthly mean values.

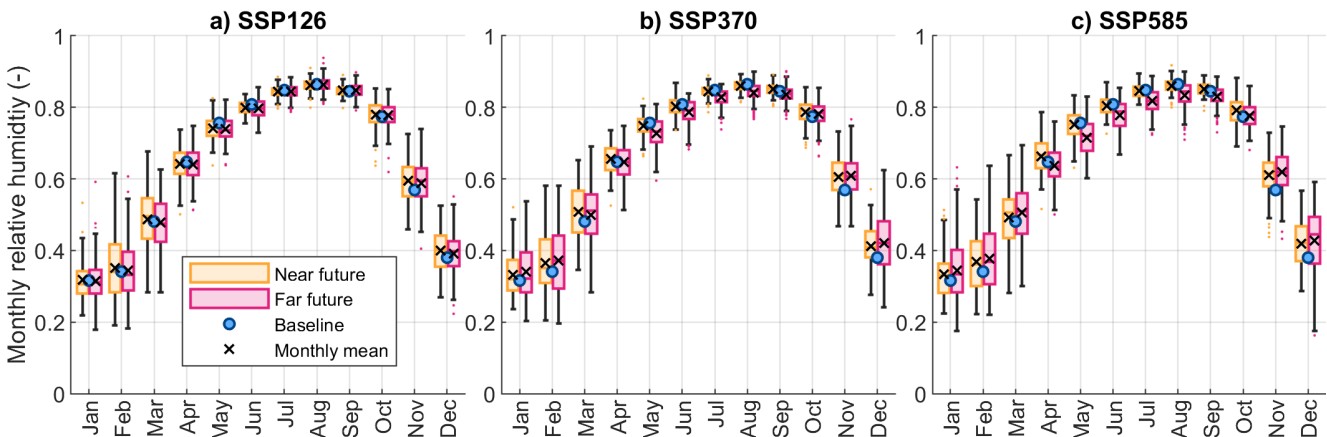

**Figure A4.** Seasonal cycle of monthly humidity projections (multi-model mean) in three climate change scenarios (SSP1–2.6, SSP3–7.0, SSP5–8.5). Blue circles represent the monthly mean humidity from ISIMIP3b baseline (1951 to 2015). Black crosses are used to highlight the multi-model monthly mean values.



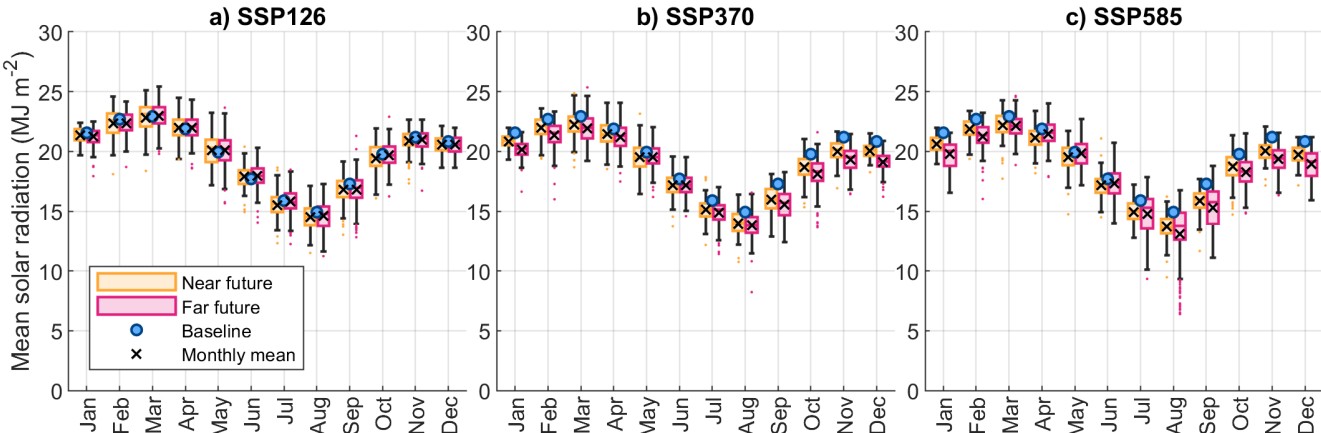

**Figure A5.** Seasonal cycle of monthly solar radiation projections (multi-model mean) in three climate change scenarios (SSP1–2.6, SSP3–7.0, SSP5–8.5). Blue circles represent the monthly mean radiation from ISIMIP3b baseline (1951 to 2015). Black crosses are used to highlight the multi-model monthly mean values.

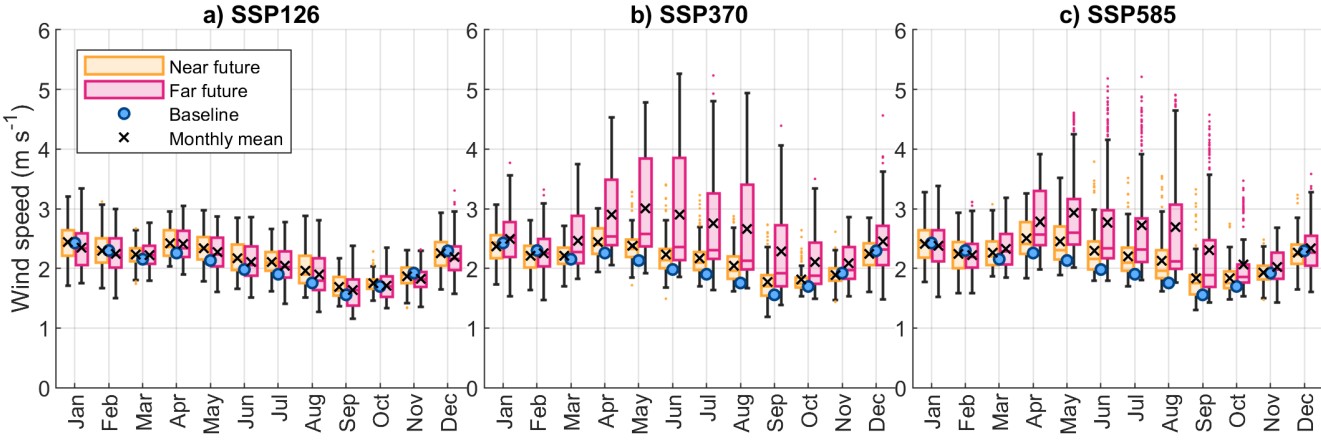

**Figure A6.** Seasonal cycle of monthly wind speed projections (multi-model mean) in three climate change scenarios (SSP1–2.6, SSP3–7.0, SSP5–8.5). Blue circles represent the monthly mean wind speed from ISIMIP3b baseline (1951 to 2015). Black crosses are used to highlight the multi-model monthly mean values.





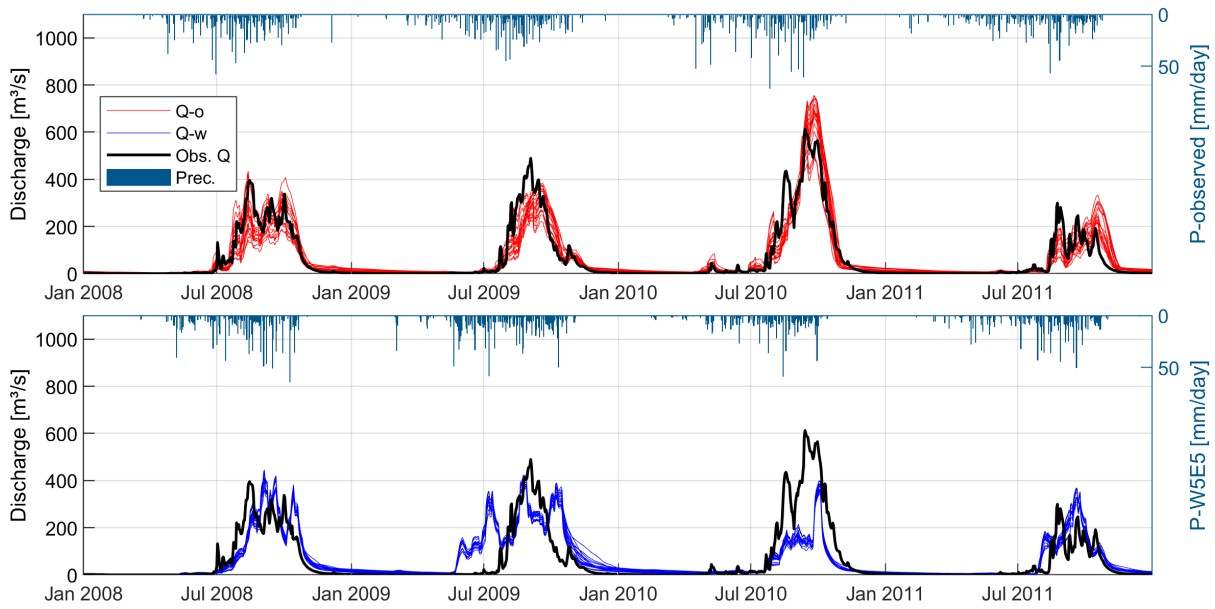

**Figure A7.** Hydrograph for the calibration period for observed forcing data (W5E5 forcing data) in the upper (lower) panel. Each panel shows the simulated hydrograph for the 20 best parameters of the "Q + LAI + AET" approach, the observed streamflow, and precipitation.

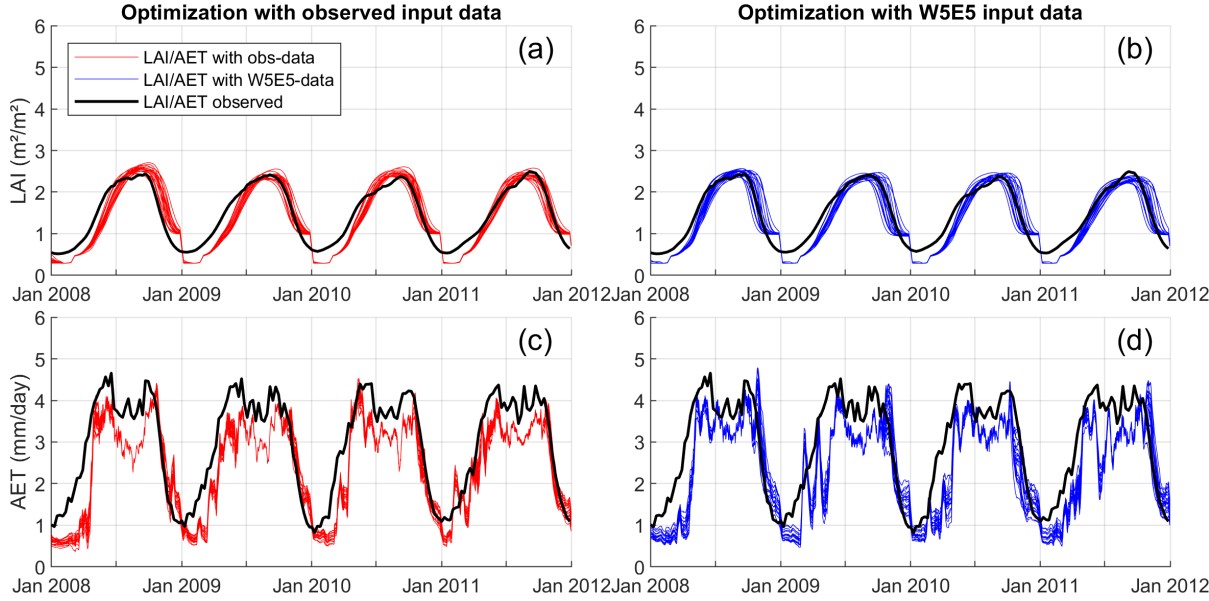

**Figure A8.** Time series of LAI/AET for the calibration period for observed forcing data (W5E5 forcing data) in the left (right) panels. Each panel shows the simulated LAI/AET for the 20 best parameters of the "Q + LAI + AET" approach and the references GLASS-LAI/FLUXCOM-AET.





**Table A1.** Projected changes of precipitation ($\Delta P$) and temperature ($\Delta T$) for the underlying GCM, emission scenario, and future period.

| GCM | SSP1–2.6 | | | | SSP3–7.0 | | | | SSP5–8.5 | | | |
|---|---|---|---|---|---|---|---|---|---|---|---|---|
| | Near | | Far | | Near | | Far | | Near | | Far | |
| | $\Delta P$ | $\Delta T$ | $\Delta P$ | $\Delta T$ | $\Delta P$ | $\Delta T$ | $\Delta P$ | $\Delta T$ | $\Delta P$ | $\Delta T$ | $\Delta P$ | $\Delta T$ |
| GFDL-ESM4 | 5.1 | 1.2 | −4.5 | 1.3 | −7.9 | 1.4 | −14.8 | 3.2 | −2.6 | 1.2 | −17.2 | 3.7 |
| IPSL-CM6A-LR | 7.9 | 1.5 | 7.6 | 1.6 | 6.6 | 1.8 | 18.4 | 4.3 | 7.0 | 1.8 | 21.3 | 5.4 |
| MPI-ESM-1-2-HR | −2.1 | 0.9 | −3.3 | 0.9 | 9.1 | 0.5 | 8.3 | 1.9 | 0.1 | 0.6 | 21.7 | 1.5 |
| MRI-ESM2-0 | 11.5 | 1.4 | −1.3 | 1.8 | 13.3 | 1.2 | −2.7 | 3.6 | 19.6 | 1.6 | −8.0 | 4.7 |
| UKESM1-0-LL | 8.7 | 1.9 | 7.4 | 2.0 | 13.2 | 2.2 | 8.9 | 4.5 | 11.2 | 1.9 | 17.8 | 5.8 |
| Multi-model mean | 6.3 | 1.4 | 1.2 | 1.5 | 7.0 | 1.4 | 3.7 | 3.5 | 7.2 | 1.4 | 7.2 | 4.2 |

**Table A2.** Overview of the mean $E_{PBIAS}$ values (in %) of the 20 parameter sets.

| Approach | Observation Network | | | | | | W5E5 dataset | | | | | |
|---|---|---|---|---|---|---|---|---|---|---|---|---|
| | Calibration | | | Validation | | | Calibration | | | Validation | | |
| | Q | LAI | AET | Q | LAI | AET | Q | LAI | AET | Q | LAI | AET |
| Q | -1.2 | 3.4 | 15.8 | -13.6 | 7.7 | 6.6 | -6.8 | 25.5 | 24.6 | -25.5 | 24.8 | 21.5 |
| Q + LAI | -2.5 | 8.7 | 20.2 | -18.2 | 11.4 | 10.3 | -8.9 | 10.8 | 20.6 | -19.6 | 11.3 | 17.3 |
| Q + LAI + AET | 0.8 | 7.2 | 17.5 | -11.1 | 10.9 | 7.6 | -3.5 | 9.6 | 18.8 | -10.4 | 10.4 | 15.7 |

**Table A3.** Overview of the mean $E_{R2}$ values (no unit) of the 20 parameter sets.

| Approach | Observation Network | | | | | | W5E5 dataset | | | | | |
|---|---|---|---|---|---|---|---|---|---|---|---|---|
| | Calibration | | | Validation | | | Calibration | | | Validation | | |
| | Q | LAI | AET | Q | LAI | AET | Q | LAI | AET | Q | LAI | AET |
| Q | 0.79 | 0.70 | 0.68 | 0.85 | 0.59 | 0.74 | 0.40 | 0.53 | 0.65 | 0.26 | 0.58 | 0.71 |
| Q + LAI | 0.73 | 0.90 | 0.82 | 0.80 | 0.81 | 0.87 | 0.40 | 0.93 | 0.73 | 0.24 | 0.89 | 0.81 |
| Q + LAI + AET | 0.75 | 0.90 | 0.82 | 0.82 | 0.80 | 0.85 | 0.38 | 0.92 | 0.73 | 0.23 | 0.88 | 0.81 |





## A3 Appendix A3 - Supplementary equations

The equations for the Kling-Gupta efficiency $E_{KGE}$, percent bias ($E_{PBIAS}$), and coefficient of determination $E_{R2}$ are shown in the following. In this, $r$ is the linear correlation between observations and simulations; $\sigma_{sim}$ and $\sigma_{obs}$ are the standard deviation of the simulations and observations, respectively; and $\mu_{sim}$ and $\mu_{obs}$ are the mean value for the simulations and observation, respectively; $n$ is the number of observations; and $Q_{sim,i}$ ($Q_{obs,i}$) is the simulated (observed) discharge at time $i$.

$$E_{KGE} = 1 - \sqrt{(r-1)^2 + \left(\frac{\sigma_{sim}}{\sigma_{obs}} - 1\right)^2 + \left(\frac{\mu_{sim}}{\mu_{obs}} - 1\right)^2}, \tag{A1}$$

$$E_{PBIAS} = 100 \times \frac{\sum_{i=1}^{n}(Q_{sim,i} - Q_{obs,i})}{\sum_{i=1}^{n} Q_{obs,i}}, \tag{A2}$$

$$E_{R2} = \left(\frac{\sum_{i=1}^{n}(Q_{obs,i} - \mu_{obs})(Q_{sim,i} - \mu_{sim})}{\sqrt{\sum_{i=1}^{n}(Q_{obs,i} - \mu_{obs})^2}\sqrt{\sum_{i=1}^{n}(Q_{sim,i} - \mu_{sim})^2}}\right)^2. \tag{A3}$$





## A4   Appendix A4 - Final parameters

**Table A4.** Final parameters for each calibration approach and forcing data. The table lists the minimal, median, and maximal parameter value from the best performing 20 sets. The superscripts denote: [1] parameter change is a multiplier, i.e., the original parameter $p$ is multiplied: $p_{new} = p_{orig} + p_{orig} \times p_{change}$, based on the parameter values from Judex and Thamm (2008); [2] parameter change is a multiplier, i.e., the original parameter $p$ is multiplied: $p_{new} = p_{orig} \times p_{change}$, based on the parameter values from Merk et al. (2024).

| Parameter | Forcing: observed data | | | Forcing: W5E5 data | | |
| --- | --- | --- | --- | --- | --- | --- |
| | Q only | Q + LAI | Q + LAI + AET | Q only | Q + LAI | Q + LAI + AET |
| ESCO | 0.16, 0.45, 0.80 | 0.28, 0.54, 0.78 | 0.14, 0.34, 0.77 | 0.28, 0.49, 0.80 | 0.19, 0.57, 0.85 | 0.25, 0.64, 0.86 |
| EPCO | 0.20, 0.40, 0.84 | 0.27, 0.42, 0.77 | 0.21, 0.47, 0.58 | 0.42, 0.56, 0.75 | 0.24, 0.40, 0.70 | 0.23, 0.43, 0.52 |
| GWQMN | 1325, 1642, 1857 | 1345, 1593, 1713 | 952, 1318, 1649 | 1012, 1197, 1514 | 1147, 1311, 1479 | 1141, 1303, 1483 |
| GW_DELAY | 1.2, 16.7, 43.3 | 6.0, 21.4, 40.1 | 7.7, 45.9, 56.2 | 5.4, 8.1, 11.0 | 8.0, 10.1, 16.0 | 5.3, 8.3, 16.3 |
| GW_REVAP | 0.04, 0.10, 0.13 | 0.04, 0.12, 0.19 | 0.07, 0.14, 0.18 | 0.07, 0.10, 0.13 | 0.07, 0.11, 0.15 | 0.08, 0.10, 0.13 |
| RCHRG_DP | 0.03, 0.12, 0.18 | 0.06, 0.11, 0.17 | 0.06, 0.15, 0.19 | 0.04, 0.07, 0.13 | 0.05, 0.11, 0.14 | 0.04, 0.07, 0.16 |
| REVAPMN | 515, 751, 955 | 399, 754, 1137 | 464, 855, 1280 | 506, 703, 1169 | 640, 788, 1132 | 671, 895, 1213 |
| SOL_AWC[1] | -0.23, 0.19, 0.44 | -0.09, 0.14, 0.28 | 0.04, 0.21, 0.45 | -0.10, 0.08, 0.34 | -0.31, -0.08, 0.33 | -0.19, 0.17, 0.40 |
| SOL_BD[1] | 0.09, 0.20, 0.32 | 0.16, 0.30, 0.38 | 0.25, 0.30, 0.32 | 0.05, 0.15, 0.28 | -0.04, 0.12, 0.26 | -0.13, 0.08, 0.29 |
| CAN_MX | 0.94, 4.47, 7.68 | 1.94, 5.72, 7.86 | 2.38, 4.39, 8.42 | 1.42, 3.56, 6.49 | 1.57, 5.37, 7.77 | 2.74, 6.36, 9.31 |
| FRGRW$_2$[2] | 0.81, 0.91, 1.04 | 0.82, 0.90, 1.00 | 0.78, 0.86, 0.96 | 0.85, 0.92, 1.07 | 0.75, 0.91, 1.03 | 0.77, 0.88, 0.98 |
| DLAI[2] | 0.77, 0.89, 1.06 | 0.86, 0.97, 1.02 | 0.87, 0.98, 1.06 | 0.80, 0.94, 1.04 | 0.83, 0.96, 1.03 | 0.83, 0.94, 1.03 |
| T_BASE[2] | 0.78, 0.97, 1.16 | 0.99, 1.10, 1.22 | 0.87, 1.04, 1.21 | 0.91, 1.02, 1.16 | 0.85, 1.00, 1.13 | 0.84, 0.94, 1.12 |
| GSI[2] | 0.84, 1.03, 1.18 | 0.82, 1.02, 1.19 | 0.81, 1.03, 1.13 | 0.86, 0.95, 1.05 | 0.88, 1.00, 1.12 | 0.84, 1.04, 1.10 |
| PHU[2] | 0.86, 0.94, 1.16 | 1.05, 1.12, 1.20 | 1.01, 1.14, 1.23 | 0.97, 1.08, 1.14 | 0.91, 0.99, 1.13 | 0.94, 1.06, 1.14 |





## A5   Appendix A5 - Water balance components

**Table A5.** Overview of the multi-model mean values of the water balance components. The baseline reference is the evaluation period 2001 to 2015 simulated with W5E5. The values show the mean value for each GCM, emission scenario, and the 20 sets of parameters in mm yr$^{-1}$. The changes to the baseline (in brackets) are percentage changes. Precipitation is the mean annual precipitation for the near- and far-future period. SWAT-T computes "Water Yield" as the sum of surface runoff, lateral flow, and baseflow. Percolation serves as proxy for groundwater recharge when using SWAT-T.

| | Near future | | | Far future | | |
|---|---|---|---|---|---|---|
| | Q | Q + LAI | Q + LAI + AET | Q | Q + LAI | Q + LAI + AET |
| **SSP1–2.6** | | | | | | |
| Precipitation | 1246 | 1246 | 1246 | 1175 | 1175 | 1175 |
| AET | 858 (+5.6) | 900 (+5.9) | 917 (+7.2) | 846 (+4.0) | 886 (+4.2) | 925 (+8.3) |
| Water Yield | 182 (−15.6) | 180 (−18.4) | 168 (−18.8) | 181 (−15.6) | 180 (−18.4) | 201 (−3.0) |
| Percolation | 221 (−6.6) | 162 (−9.2) | 158 (−9.8) | 185 (−22.0) | 130 (−26.7) | 154 (−12.0) |
| **SSP3–7.0** | | | | | | |
| Precipitation | 1238 | 1238 | 1238 | 1203 | 1203 | 1203 |
| AET | 860 (+5.7) | 898 (+5.8) | 915 (+7.1) | 880 (+8.2) | 918 (+8.0) | 932 (+9.1) |
| Water Yield | 180 (−16.7) | 177 (−19.6) | 167 (−19.2) | 179 (−16.8) | 177 (−19.6) | 167 (−19.2) |
| Percolation | 216 (−8.6) | 160 (−10.1) | 157 (−10.3) | 180 (−23.9) | 127 (−28.3) | 124 (−29.4) |
| **SSP5–8.5** | | | | | | |
| Precipitation | 1241 | 1241 | 1241 | 1244 | 1244 | 1244 |
| AET | 847 (+4.2) | 888 (+4.5) | 904 (+5.7) | 889 (+9.3) | 911 (+7.2) | 926 (+8.3) |
| Water Yield | 216 (+0.5) | 213 (−3.0) | 200 (−3.0) | 217 (+0.5) | 213 (−2.9) | 201 (−3.0) |
| Percolation | 227 (−4.4) | 169 (−5.2) | 166 (−5.1) | 201 (−15.1) | 157 (−11.8) | 153 (−12.0) |

## A6   Appendix A6 - References of the study site map

**Table A6.** Overview of the data use for the study site map.

| Data | Database name or source |
|---|---|
| Topography | Copernicus GLO-30 (Copernicus, 2022) |
| Land use map | Copernicus Global Land Service (Buchhorn et al., 2020) |
| Water bodies | ArcGIS Pro 2.7.3 (ESRI) |
| Countries and cities | ArcGIS Pro 2.7.3 (ESRI) |
| Catchment extents | Derived with ArcGIS Pro 2.7.3 (ESRI) |



*Author contributions.* FM, TS, FA, MR, JB, MD reviewed and edited the manuscript; FM, TS, FA, MD conceived and designed the study; FM, TS, FA, MR, JB acquired the data; FM, TS, FA performed the data analysis and model development and simulations; FM, TS, FA, MR, JB, MD evaluated the simulations and models. FM wrote the manuscript draft.

*Competing interests.* The authors declare that they have no conflict of interest.

*Acknowledgements.* The authors want to thank the BMBF ("Bundesministerium für Bildung und Forschung") for the funding of the FURI-FLOOD research project ("Current and future risks of urban and rural flooding in West Africa", Grant No.: 01LG2086B). We would like to thank our partners involved in the FURIFLOOD project for their support. This work was further supported by the European Union's Horizon Europe research and innovation program as part of the UAWOS project ("Unmanned Airborne Water Observing System", Grant Agreement
No.: 101081783).



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
