# Peer review of "The impact of calibration strategies on future evapotranspiration projections: a SWAT-T comparison of three hydrological modeling approaches in West Africa"

_EGUsphere, 2025_

## Author Comment (AC1)

| Comment | Response |
|---|---|
| Reliance on MODIS-based products (GLASS-LAI and FLUXCOM AET)

• Although you note the use of GLASS-LAI and FLUXCOM, both high-quality datasets, there is no discussion of the uncertainties associated with remotely sensed products, which I think should be acknowledged. | We thank the reviewer for this important suggestion. This referee comment aligns with the feedback from Referee #2 in [https://doi.org/10.5194/egusphere-2025-3836-RC1]. We therefore respond similar to both referee comments:

GLASS-LAI and FLUXCOM-AET are both widely used and are considered high quality. In tropical regions, remote sensing can be influenced due to cloud cover. The cloud influence is particularly prominent in MODIS-derived LAI products. Therefore, we used the more robust GLASS-LAI product. The FLUXCOM product integrated eddy covariance observations, yet these are unevenly distributed across climate zones.

As mentioned before, a comparison of GLASS-LAI to modeled LAI with SWAT-T can be found in Merk et al. (2024). We tested the FLUXCOM-AET accordingly before its application in this study, see figure below. Given this promising validation, the products still carry inherent uncertainties.
We have added a paragraph discussing the limitations and uncertainties with GLASS-LAI and FLUXCOM-AET.
We now explicitly acknowledge that these uncertainties may influence the representation of LAI and AET at the sub-basin scale. Including this discussion clarifies the scope of our validation and improves transparency regarding data limitations.

The following paragraph is added to the manuscript after line 502:
*The application of satellite-based GLASS-LAI and FLUXCOM-AET data for a tropical catchment like the Bétérou Catchment carries uncertainties. Generally, GLASS-LAI and FLUXCOM-AET are both widely used and are considered high quality. Yet, satellite-based datasets in tropical regions can be subject to cloud contamination and reflectance noise (Viovy et al., 1992; Atkinson et al., 2012) or the lack of observation networks for validation (Weerasinghe et al., 2020). For the Bétérou region, the GLASS-LAI dataset shows promising applicability when compared to monitored data (Merk et al., 2024). FLUXCOM-AET has been extensively validated against eddy covariance (EC) measurements across a wide range of climatic conditions, supporting its general reliability (Jung et al., 2019). This study relies on a pointwise validation of FLUXCOM-AET and GLASS-LAI. While this approach does not fully capture spatial heterogeneity or all sources of uncertainty, it provides a consistent and pragmatic basis for model evaluation in data-scarce tropical regions. This approach ensures that the validation framework remains robust and fit for purpose.* |

| | |
|---|---|
| You comment throughout that you have determined LAI is a good proxy for AET, however I find this a challenging takeaway that is not entirely proven. For example:

• Currently, there is no calibration that uses Q + AET only (QA). It would be useful to either include this configuration or explain why it wasn't done and whether it would meaningfully change the results. I imagine it would change the results, as at the moment, the QL and QLA calibrations behave similarly. Could this simply be because LAI is driving the optimisation? Showing the QA calibration (or explaining its omission) would help clarify this. If QA performs similarly to QL, that would support the idea that LAI could be a proxy for AET. | We appreciate the referee's comment on the methodology and calibration approaches. The following explanation briefly summarizes the approach definition and selection.

In the region, LAI and AET are strongly linked to each other as shown in literature and observational data. The overarching objective of the study is to evaluate the implications of calibration strategies to estimate AET in a sub-humid and AET-dominant catchment. In addition, the study discusses whether detailed LAI modeling can serve as proxy for predicting AET, even if AET is disregarded in the calibration.

For this purpose, three calibration approaches are defined. The Q-only strategy neglects LAI and AET in the optimization and is used as a baseline to understand the AET modeling performance if it is not integrated at all. The Q-only strategy serves as a lower limit in the approach comparison.
The Q+LAI strategy includes LAI, but neglects AET in the calibration. With this approach, it can be evaluated how LAI influences the AET estimation.
With the application of multi-objective calibration, hydrological models are more robust. The Q+LAI+AET strategy includes all three variables in the optimization. It serves as an upper limit to evaluate best possible optimization with respect to Q, LAI, and AET.

The Q+AET strategy would be a potential upper limit for the comparison just like Q+LAI+AET. It can be expected that Q+AET represents well AET rates but so does Q+LAI+AET. From Q+AET calibration, it can be expected that vegetation modeling is neglected to satisfy the water demands of AET.

The extent of the study is extensive. As an upper limit in the strategy comparison, we defined a model setup which is representative for the region (Q, LAI, AET). We therefore chose the Q+LAI+AET approach rather than the Q+AET for the comparison of the strategies. Still, the discussion about the upper limit for the LAI-AET evaluation at the catchment scale is crucial. To account for this relevant point, we added a paragraph to the discussion section in the revised manuscript (line 494 and following):

*The investigation of a Q+AET calibration could also serve as an upper benchmark and is expected to reproduce AET rates well. It is however not considered in the present study, because it would potentially neglect vegetation dynamics to satisfy the atmospheric water demands. Isolating Q+AET from LAI adds a further layer in understanding the role of vegetation processes in AET-focused calibrations. Looking ahead, future work could explicitly explore Q+AET as a further upper limit together with Q+LAI+AET.* |

| | |
|---|---|
| • Further, the KGE values improve for both AET and streamflow when AET is included (QLA compared to just QL). This raises the question of whether LAI alone adds enough information. | We thank the reviewer for this valuable observation. The study evaluates the impact of different calibration strategies on AET estimation and particularly discusses the role of LAI for AET modeling. AET is an optimization target in the Q+LAI+AET approach, but its optimization is neglected in Q+LAI. Therefore, better AET representation and higher KGE values can be expected for Q+LAI+AET than Q+LAI. For streamflow, the influence of LAI is less significant. The significance of parameters and processes on the streamflow modeling with SWAT-T can be evaluated through the Morris analysis (Figure 3). In Figure 3c, the sensitivity with respect to streamflow is shown where LAI parameters (green circles) are less important than groundwater (orange circles) or AET (purple circles). Therefore, LAI is less sensitive for streamflow, but it is important for AET (Figure 3d). Please note that the caption in Figure 3 will be adjusted in the revised manuscript: panel 3d is "$E_{KGE,AET}$" and 3c is "$E_{KGE,Q}$".

 We adjusted the manuscript for the caption of Figure 3 and added that LAI alone does not necessarily improve the streamflow modeling with SWAT-T. The manuscript is adjusted in line 389:
 *The analysis in Figure 5 further indicates that the integration of LAI alone does not necessarily improve the streamflow modeling with SWAT-T. As shown in Figure 3, the influence of LAI parameters on streamflow prediction is less sensitive.* |
| • Additionally, as noted in your study, none of the calibration strategies reach the benchmark KGE value suggested by Knoben et al. (2020). It may be worth discussing whether LAI limits performance, and whether AET alone could achieve closer to benchmark values. | We highly appreciate the referee's comment on the benchmark limits. We assume that LAI-only and AET-only approaches can increase the KGE values. However, we also expect the LAI-only and AET-only approach to not beat the seasonal benchmarks. GLASS-LAI and FLUXCOM-AET are re-analysis products based on data assimilation and are generated based on seasonal dynamics and statistical means. While they serve as a reliable reference for land-atmosphere modeling, it is challenging for process-based models to outperform those datasets. Although the integration of AET-only and LAI-only is beneficial to assess the model performance against the benchmarks, we focus the analysis on the three defined approaches to keep the manuscript concise and coherent with the climate impact assessment. |
| Suitability of KGE for LAI

 • KGE is an appropriate metric for streamflow and for AET given the daily FLUXCOM data. However, LAI changes slowly and has limited intra-seasonal variability, so I question whether KGE is the most informative metric for LAI. I | In total, three performance criteria are used to evaluate Q, LAI, and AET: KGE, PBIAS and $R^2$. The usage of the performance metrics is explained in Chapter 2.5, line 277. We focus the model optimization on KGE because the KGE metrics considers three statistical means simultaneously: the linear correlation between observation and simulation; the standard deviations as a measure of variability; and the mean values as a measure of bias. The application of KGE is therefore |

| | |
|---|---|
| suggest either using a LAI-specific metric (e.g., RMSE, bias, seasonal amplitude/timing) or including a justification for using KGE in the manuscript. | a comprehensive way to include different statistical means into the performance quantification.

The combination of KGE with the percent bias (PBIAS, see Table A2 in appendix) and $R^2$ (see Table A3 in appendix) further enables a comprehensive evaluation of the LAI dynamics. |
| Figures 9 and 10

• These are difficult to interpret without a baseline figure showing monthly AET in this format. Consider instead presenting percentage change in monthly AET, which may communicate the intended comparison more clearly, or showing a baseline figure (even in the appendix). | Thank you for pointing out that Figures 9 and 10 are unclear to the reader. The figures are intended to show the baseline monthly AET for each strategy. The baseline for each month and strategy is indicated through the black crosses.

We adjusted the caption to make it clearer that Figures 9 and 10 also contain the baseline monthly values. |
| Methods section

• The Methods read a little cluttered in parts, distracting from the overall story. You could consider moving some elements (e.g., the sensitivity analysis description) to the supplementary materials, to tighten this. | We highly appreciate the reviewer's feedback on the phrasing and structure of the Methods section.

We moved the equation parts of the description of the sensitivity analysis to the supplementary material and improved the cluttering and phrasing of the section to improve the readability for the reader. |

**Technical Corrections**

| | |
|---|---|
| Line 72 – "if or if not" is awkward. Suggest "whether" or "whether or not." | We highly appreciate the reviewer's valuable feedback on language, wording, and phrasing to improve the readability and overall quality of the manuscript. |
| Line 140 – "We use the Penman–Monteith method..." appears to be repeated. | We adjusted the manuscript for the suggested technical corrections. |
| Line 189 – Extra space before the bracket in "(5 km resolution)." | |
| Figure 3 – Consider adding detail to panel lettering (e.g., "a) all variables", "b) LAI..."). Increase font size of u* on the axes and briefly define it in the caption (e.g., "higher u* = more sensitive"). | |
| Figure 5 – Consider relabelling the y-axis to something clearer (e.g., "Cumulative probability") or define F(x) in the caption. | |
| Line 398 – missing "to" after according. | |
| Figure 8 – Consider adding the projection period to each panel | |

| | |
|---|---|
| (e.g., "a) 2031–2050" or "a) near-future"). | |
| Figure 11 – Axis font sizes are too small; consider increasing. | |
| Lines 489–490 – The sentence structure is unclear, and "with particularly for AET" is incorrect grammar. A clearer option might be: "Similar to previous comparative studies, we investigate simple to comprehensive calibration strategies, with a particular focus on AET." | |

**Supplemental Figures**

[Figure]

**Figure 1** Comparison of monitored AET (AET-obs, black line) to multiple satellite-based and reanalysis AET products at the forested footprint for different temporal scales a) daily, b) 8-daily sum, and c) monthly sum. Please note the varying y-axis scales for readability.

**Literature**

Please note that literature already cited in the manuscript is not added to the following list.

Atkinson, P.M., Jeganathan, C., Dash, J., Atzberger, C., 2012. Inter-comparison of four models for smoothing satellite sensor time-series data to estimate vegetation phenology. Remote Sensing of Environment 123, 400–417. doi:10.1016/j.rse.2012.04.001

Viovy, N., Arino, O., Belward, A.S., 1992. The best index slope extraction ( bise): A method for reducing noise in ndvi time-series. International Journal of Remote Sensing 13, 1585–1590. doi:10.1080/01431169208904212.

Weerasinghe, I., Bastiaanssen, W., Mul, M., Jia, L., and van Griensven, A.: Can we trust remote sensing evapotranspiration products over Africa?, Hydrol. Earth Syst. Sci., 24, 1565–1586, https://doi.org/10.5194/hess-24-1565-2020, 2020

---

## Author Comment (AC2)

| Comment | Response |
|---|---|
| Line 15–20

You state that "the combined 'Q + LAI' can be used as a proxy to predict AET rates." This is an important claim but needs further qualification. While your results suggest that LAI contributes to improved AET estimates, it is not fully demonstrated that this proxy relationship holds across climatic variability or land use types. You may consider referencing doi: [10.1016/j.scitotenv.2020.143792], which evaluates how dynamic LULC influences surface runoff and can affect AET estimates as well. | The study covers the analysis of two different meteorological forcing data sets, three calibration approaches, and 20 best-fit parameter sets per modeling approach. Therefore, simulation results are extensive. Moreover, the spatial resolution of the satellite-based LAI and AET data is 250x250 m and 0.0833°x0.0833° (ca. 9x9 km). Comparing satellite-based and simulated LAI and AET, the evaluation is carried out at the subbasin scale. To summarize the evaluation, we focus on a subbasin- and catchment scale evaluation with area-weighted performance criteria (here Kling-Gupta efficiency). At these scales, a clear improvement between the approaches Q-only and Q+LAI can be observed.

Notably, smaller scales (e.g., HRU levels) and dynamic land use changes have not been considered given the spatial mismatch of modeled and simulated LAI and AET. An assignment of a satellite-based AET pixel to SWAT-T's HRU scale is challenging because of the different spatial resolutions.

The proposed study is an excellent example of how LULC impacts AET estimations. We will highlight in the revised manuscript the limitations that no explicit LULC dynamics have been considered and include the mentioned study as a prime example of why integration is important.

In the revised version, we have adjusted the claim already in the abstract. More precisely, we highlight that our findings for LAI and AET focus on subbasin- and catchment scale (line 15-20):
*The results moreover demonstrate that the combined "Q + LAI" can be used as a proxy to predict AET rates at the catchment scale.*

The proposed study has been added to the discussion part (line 496):
*Smaller scales (e.g., HRU levels) and dynamic land use changes have not been considered given the spatial mismatch of modeled and simulated LAI and AET. As demonstrated in Yonaba et al. (2021), dynamic land use modeling can further enhance the process representation of surface runoff and AET in the region. Future research can consider a coupling of detailed LAI, AET, and dynamic land use modeling to further enhance robust hydrological modeling for the region.* |
| Lines 28–30

The discussion of AET as an "essential role in the regional hydrology" is well-stated but would benefit from a more precise articulation of the mechanisms, | The major objective of the study is to discuss the relevance of AET calibration approaches. Of course, the seasonal feedbacks as well as the soil moisture are paramount for the regional hydrology. We agree that the statement is "too short".

We added the relevance of soil moisture to the statement in the revised manuscript: |

| | |
|---|---|
| e.g., the seasonal feedbacks between soil moisture, vegetation phenology, and transpiration. Currently, the sentence reads too broadly. | *In the regional hydrology in the sub-humid parts of West Africa, AET, vegetation phenology, and soil moisture are paramount to the land-atmosphere feedbacks (Hingerl et al., 2025). In these mechanisms, AET plays an essential role where its rate to precipitation can be up to 80 % (Rodell et al., 2015).* |
| Lines 61–70

 This section criticizes prior studies for omitting AET in model calibration. However, the critique would be stronger if you quantified the extent of the bias introduced by using only discharge. For example, what is the typical error in AET projections under Q-only calibration across your GCM ensemble? Additionally, reference could be made to the limitations of using discharge-only calibration in semi-arid systems, as shown in doi: [10.3390/land12112017]. | We revised the section and added the numbers to make the critique stronger. In addition, we added the proposed study to give the reader an example of a close region to further discuss the limitations of Q-only approaches.

 We will revise the section (lines 65-70):
 *Most others estimate future AET from SWAT models being optimized with discharge-only calibration approaches (Bossa et al., 2012; Sood et al., 2013; Bossa et al., 2014; Danvi et al., 2018; Larbi et al., 2021; Animashaun et al., 2023, **Zouré et al., 2023**). However, the modeling of AET in process-describing hydrological models, such as SWAT/SWAT+, substantially relies on parameters that also influence runoff generation, e.g., interception or soil moisture. **As shown by our study in the following, calibration approaches without AET calibration result in a future AET increase of 4.0% compared to an increase of 9.2% when AET is integrated in the model optimization.** Hence, without explicit model calibration against AET, projections of future AET can remain constrained and potentially misleading.* |
| Lines 88–90

 The assertion that this study "contributes to minimize uncertainties from model calibration approaches" should be reworded. Rather than claiming uncertainty minimization, a more accurate statement might be that the study demonstrates how calibration strategies affect model spread and projections. | We thank the referee for this important feedback. The statement is worded in a simplified way and clarification certainly improves the quality of the manuscript.

 In the revised version, we avoid the statement about uncertainty minimization and focus on a more accurate statement. The section is adjusted to (line 85):
 *This study demonstrates the implications of calibration strategies on the model spread and projection for AET estimation.* |
| Lines 104–108

 The explanation of SWAT-T improvements over SWAT is helpful, but further clarification is needed on how the tropical phenology routines specifically alter AET dynamics. For instance, how does the model account for year-round biomass retention or multi-modal LAI cycles? Please consider adding a brief example or case insight. | In the SWAT-T model, different states of vegetation growth are represented. The start of the plant growth from dormancy is triggered by soil moisture changes in the shift from dry to wet period, plant maturity is modeled in the wet season, and leaf senescence as well as vegetation dormancy are accounted for. As an example, we added Figure A8 to the Appendix of the article. The figure shows the weighted average of LAI across the catchment.

 The manuscript is lengthy, and the focus is on the implications of the calibration approaches. We have highlighted the case study of Alemayehu et al. (2017) and refered to their study for the reader for a detailed plant phenology with SWAT-T. |

| | We adjusted the section in the manuscript: |
|---|---|
| | *The key enhancement of SWAT-T is the modification of the plant growth module to better represent the plant phenology in tropical regions (Alemayehu et al., 2017).* **A detailed explanation of the advances as well as a comprehensive case study of SWAT-T is presented in Alemayehu et al. (2017).** *In brief, the SWAT-T model leads to more accurate predictions of AET in regions with perennial plant growth (Zhang et al., 2020; Fernandez-Palomino et al., 2021; Ferreira et al., 2021; López López et al., 2017; Merk et al., 2024).* |
| Lines 144–154

The plant parameters were fixed based on prior literature, but it is unclear whether those parameter values are transferable across different land covers or years. Given that some of these parameters (e.g., LAIMX2, PHU) can be sensitive to local agronomic practices and interannual climate variability, this approach could bias calibration. Have you evaluated the robustness of this transfer? Could a partial re-optimization for these parameters improve model skill? | We applied a sensitivity analysis (Morris method) to identify the most relevant parameters for Q, LAI, and AET modeling. We fixed the less-sensitive parameters to decrease the parameter space.

We checked whether the transfer from literature is appropriate and observed promising fits for LAI across the three main land cover types. The transferred parameters were derived in region which is located within the Bétérou Catchment (Merk et al., 2024). Given this spatial proximity, the transfer is applicable. Further calibration can improve the model's performance. Still, our modeling approach has shown strong agreement of simulated and satellite-based LAI.

However, the transfer is not necessarily applicable to other regions, given different climates or agronomic practices. We have not checked the robustness of the fixed parameters. The transfer of LAI parameters can not be guaranteed due to regional properties, such as climate, vegetation type, and agronomic practices. |
| Lines 165–175

The land use representation for croplands through "AGRL" seems overly simplified. Since croplands cover a substantial portion of the catchment, this may limit the realism of AET dynamics, particularly in peak growing periods. Have you considered applying crop-specific growth curves or incorporating a dynamic planting calendar? | We chose the "AGRL" representation for crop fields due to two reasons. First, we had no access to the spatial distribution of the crops in the catchment. We relied on the cropping calendar presented by Forkour et al. (2014). They published a cropping calendar consisting of the six most prominent crops in the area. Based on this calendar, we derived an "average" crop representation. We assigned the "AGRL" crop type from the SWAT data base, but adjusted the corresponding LAI parameters in the plant.dat-file.
Second, we evaluated the "average" crop against satellite-based LAI and AET prior the model optimization. We observed a good agreement of the "average" crop to the reference.

Thus, we argue that while the crop representation is simplified, it is still appropriate and applicable. For areas with higher cropland share, like the lower parts of the Ouémé river in Benin, this approach may not be applicable. |
| Lines 195–205

The aggregation of GLASS-LAI and | The aggregation is necessary because of the spatial mismatch of GLASS-LAI and FLUXCOM-AET and the simulated model outputs. As gridded datasets, GLASS-LAI and FLUXCOM-AET |

| | |
|---|---|
| FLUXCOM-AET data to the subbasin scale is described clearly. However, you should mention potential scale mismatches between remote sensing and model HRUs. This aggregation likely introduces smoothing effects that may obscure localized land-atmosphere feedbacks. Please discuss whether this mismatch may influence LAI-AET correlation strength. | inherently represent mean conditions over each pixel. Consequently, fine-scale variability and localized land–atmosphere processes are smoothed in the aggregated values.

Each dataset has been validated for different regions. As a matter of fact, local land-atmosphere processes are smoothed in this dataset. Many hydrological questions require us to upscale us to larger spatial domains (e.g., catchment scale) and in this context the "smoothed" representation provided by GLASS-LAI and FLUXCOM-AET remains appropriate for assessing broader patterns, even though localized feedbacks may be attenuated.

A comparison of GLASS-LAI to modeled LAI with SWAT-T can be found in Merk et al. (2024). We tested the FLUXCOM-AET accordingly before its application in this study, see Figure 1 in the *Supplemental Figures* section. These comparisons show that, while some smoothing is unavoidable, the aggregated datasets are still suitable for examining LAI-AET relationships at the sub-basin scale.

In the discussion part, we discuss the aggregation of LAI-GLASS and FLUXCOM-AET. We further adjusted the manuscript (line 499 and following):
*While this approach allows for a spatially explicit evaluation that reflects land cover variability within the Bétérou catchment, it also introduces some information loss, e.g., from the aggregation of AET heterogeneity.* **This is particularly important in the process representation of the LAI-AET interaction for localized land-atmosphere feedbacks.** *The subbasin-level evaluation offers a practical compromise, where spatial context is preserved at a scale appropriate for hydrological modeling.* |
| Lines 225–235

The explanation of the Morris method is mathematically accurate, but its description interrupts the methodological flow. Consider moving equations to the Supplementary Material or shortening the derivation here. Focus instead on the reasoning for using Morris over Sobol or other variance-based methods. | Thank you for this recommendation.
We moved the equation part to the Supplementary Material and highlighted the rationale behind the choice for the Morris method.

The following paragraph has been added to the manuscript:
*The application of the Morris method allows the quantification of the parameter sensitivity through statistical means ($\mu^*$-values) of the elementary effects (Morris, 1991; Campolongo et al., 2007). While the elementary effects are a local sensitivity measure, the computation of multiple elementary effects for varying parameter subsets enables a quasi-global sensitivity analysis (Saltelli et al., 2008). Compared to other sensitivity approaches, it is moreover computationally efficient due to its one-at-a-time sampling strategy (Pianosi et al., 2016).* |
| Lines 254–266

The role of half-space depth in ROPE is well explained, but please | The convergence was assessed in a simplified approach. We defined the convergence if the KGE of the best run to the last run does not change more than ΔKGE=0.05. To be on the safe side, we defined enough iterations rather than a stopping criterion. |

| | |
|---|---|
| clarify how convergence was assessed. Did you apply any stopping criteria based on performance plateauing? Were all final parameter sets within the convex hull? Details like these improve transparency and reproducibility. | This is computational costly but given 15 parameters and 3 optimization objectives (Q, LAI, AET), we wanted to allow enough time for the convergence to be met.
The ROPE algorithm defines the samples for the next iteration based on a depth function (half space depth function). This depth function requires a high computational demand for 15 parameters yet is robust to sample within the convex hull. All parameter sets are within the convex hull. |
| Lines 300–310

You use the W5E5 simulation period (2001–2015) as a baseline. However, earlier you mentioned that observed data go back to 1981. Why was a shorter baseline chosen, especially given that longer baselines can reduce noise in climate signal detection? | We appreciate the reviewer's observation. We selected the period 2001 to 2015 as a baseline to ensure consistency across all data sets used in the study. In the 15 years span, the interannual climate variability is well captured and no significant climate change trend is observed. In the preprocessing of the data, no significant noise was further detected. |
| Lines 361–375

The seasonal patterns of AET are well captured in the model, particularly the mid-season dip. However, your explanation that this dip is "due to lack of LAI representation" in Q-only is somewhat superficial. Could the effect also arise from misrepresented soil moisture dynamics or canopy interception? | We also evaluated the soil moisture response for the simulation of the current climate conditions, see for example Figure 2.

The different modeling strategies simulate similar patterns of soil moisture. More precisely, the soil moisture dynamics are similar among the strategies since all of them are calibrated against discharge which has the most influence on soil moisture in the SWAT/SWAT-T model.

Canopy interception $E_{can}$ is linked to LAI in SWAT/SWAT-T with the formula:
$$E_{can} = I_{max} * \frac{LAI}{LAI_{max}}$$
Where $I_{max}$ is the maximum canopy interception (model parameter, CAN_MX), LAI is the computed LAI, and $LAI_{max}$ is the maximum LAI (model parameter, BLAI). Therefore, inappropriate modeling of LAI (e.g, very small values for LAI) influences canopy interception.

Given the soil moisture is similar among the strategies and canopy interception is linked to LAI, we concluded that the "AET-dip" representation is strongly due to limits in LAI modeling. |
| Lines 395–405

The cross-validation results show that W5E5-derived parameters can be used with observed forcing. However, the validation for LAI shows noticeable degradation. This suggests that model structural limitations may constrain LAI robustness under | We appreciate the reviewer's observation and the important aspect of LAI it points out.
LAI modeling in SWAT/SWAT-T is linked to temperature. The LAI parameters T_BASE and T_OPT govern the temperature stress on plant growth. Moreover, T_BASE parameter influences the heat units necessary for plant growth.

Given the dependency of LAI on temperature, the robustness for climate impact assessment is crucial. The manuscript is adjusted to (line 404): |

| different climates. Could you expand on the implications of this for future climate scenario modeling? | *Given the dependency of LAI on temperature in SWAT-T, its robustness for climate impact assessment is crucial. For climate impact assessment, multiple LAI parameter sets can be used to avoid the reliance on single LAI parameter sets and increase the LAI parameter robustness.* |
|---|---|
| Lines 425–435

You evaluate GCM ensemble spread and precipitation changes, but the connection between these meteorological changes and hydrological sensitivity is underexplored. For example, how does GCM precipitation variability translate into spread in Q or AET for the different calibration approaches? A more integrated uncertainty decomposition would be valuable here. | Thank you for this important observation. The used GCM's from ISIMIP represent a range of low to high climate sensitivity.

We added an additional analysis (Table 1 and Table 2) to the manuscript to evaluate how meteorological changes from GCMs translate into changes for Q and AET. This way, we hope to improve the quality of the evaluation.

The tables are added to the *Supplemental Tables* section below to the appendix of the manuscript. |
| Throughout, there is a lack of critical reflection on the limitations of remote sensing products used for LAI and AET. While GLASS-LAI and FLUXCOM are high-quality datasets, they carry uncertainties, especially in tropical canopies. A brief paragraph discussing these limitations would strengthen the credibility of your validation approach. | We thank the reviewer for this important suggestion. This referee comment aligns with the feedback from Referee #2 in [https://doi.org/10.5194/egusphere-2025-3836-RC2]. We therefore respond similar to both referee comments:

GLASS-LAI and FLUXCOM-AET are both widely used and are considered high quality. In tropical regions, remote sensing can be influenced due to cloud cover. The cloud influence is particularly prominent in MODIS-derived LAI products. Therefore, we used the more robust GLASS-LAI product. The FLUXCOM product integrated eddy covariance observations, yet these are unevenly distributed across climate zones.

As mentioned before, a comparison of GLASS-LAI to modeled LAI with SWAT-T can be found in Merk et al. (2024). We tested the FLUXCOM-AET accordingly before its application in this study, see figure below. Given this promising validation, the products still carry inherent uncertainties.
We have added a paragraph discussing the limitations and uncertainties with GLASS-LAI and FLUXCOM-AET.
We now explicitly acknowledge that these uncertainties may influence the representation of LAI and AET at the sub-basin scale. Including this discussion clarifies the scope of our validation and improves transparency regarding data limitations.

The following paragraph is added to the manuscript after line 502:
*The application of satellite-based GLASS-LAI and FLUXCOM-AET data for a tropical catchment like the Bétérou Catchment carries uncertainties. Generally, GLASS-LAI and FLUXCOM-AET are both widely used and are considered high quality. Yet, satellite-based datasets in tropical regions can be subject to* |

| | |
|---|---|
| | *cloud contamination and reflectance noise (Viovy et al., 1992; Atkinson et al., 2012) or the lack of observation networks for validation (Weerasinghe et al., 2020). For the Bétérou region, the GLASS-LAI dataset shows promising applicability when compared to monitored data (Merk et al., 2024). FLUXCOM-AET has been extensively validated against eddy covariance (EC) measurements across a wide range of climatic conditions, supporting its general reliability (Jung et al., 2019). This study relies on a pointwise validation of FLUXCOM-AET and GLASS-LAI. While this approach does not fully capture spatial heterogeneity or all sources of uncertainty, it provides a consistent and pragmatic basis for model evaluation in data-scarce tropical regions. This approach ensures that the validation framework remains robust and fit for purpose.* |
| Finally, regarding language and terminology throughout the manuscript, I believe that phrases such as "to guarantee convergence," "mimics well," and "modeling potential" could be rephrased to enhance precision. For instance, "the model performance plateaus after 12 iterations" is preferable to "to guarantee convergence." Similarly, instead of "mimics well," you could say "closely reproduces." | We highly appreciate the reviewer's feedback on language and wording. We screened through the manuscript and adjusted poor phrasing to improve the readability. |

**Supplemental Figures**

[Figure]

**Figure 1** Comparison of monitored AET (AET-obs, black line) to multiple satellite-based and reanalysis AET products at the forested footprint for different temporal scales a) daily, b) 8-daily sum, and c) monthly sum. Please note the varying y-axis scales for readability.

[Figure]

**Figure 2** Average soil moisture for the 14 subbasins (letters a-n) for model forcing with obs-data.

**Supplemental Tables**

**Table 1** Meteorological (ΔP for precipitation change in percent; ΔT for temperature change in degree Celsius) and hydrological changes (ΔQ and ΔE for discharge and AET changes in percent, respectively) for the GCMs and SSP scenarios (short names for GCMs and SSPs are used for layout purposes; MMM = multi-model mean) and the underlying calibration strategies (1=Q-only; 2=Q+LAI; 3=Q+LAI+AET) for the near-future period.

| GCM | SSP | ΔP | ΔT | ΔQ1 | ΔQ2 | ΔQ3 | ΔE1 | ΔE2 | ΔE3 |
|---|---|---|---|---|---|---|---|---|---|
| GFDL | SSP1 | 5.1 | 1.2 | -3.2 | -6.8 | -9.0 | 2.3 | 2.8 | 4.4 |
| | SSP3 | -7.9 | 1.4 | -51.2 | -49.9 | -51.8 | 1.6 | 1.1 | 2.6 |
| | SSP5 | -2.6 | 1.2 | -30.9 | -30.1 | -32.1 | 0.9 | 0.7 | 2.2 |
| IPSL | SSP1 | 7.9 | 1.5 | -6.5 | -9.8 | -11.6 | 8.1 | 8.0 | 9.5 |
| | SSP3 | 6.6 | 1.8 | -13.4 | -16.9 | -17.7 | 10.9 | 11.3 | 12.6 |
| | SSP5 | 7.0 | 1.8 | -11.7 | -14.9 | -16.6 | 8.4 | 9.2 | 10.7 |
| MPI | SSP1 | -2.1 | 0.9 | -31.1 | -29.0 | -31.2 | 2.1 | 1.5 | 3.3 |
| | SSP3 | 9.1 | 0.5 | 7.0 | 1.3 | -2.6 | 1.6 | 1.0 | 2.7 |
| | SSP5 | 0.1 | 0.6 | -12.0 | -12.7 | -14.5 | -1.5 | -2.0 | -0.6 |
| MRI | SSP1 | 11.5 | 1.4 | 21.4 | 11.1 | 9.8 | 5.9 | 6.7 | 8.1 |
| | SSP3 | 13.3 | 1.2 | 26.7 | 13.1 | 8.8 | 4.2 | 4.9 | 6.3 |
| | SSP5 | 19.6 | 1.6 | 48.1 | 27.5 | 22.5 | 4.1 | 5.1 | 6.4 |
| UKESM | SSP1 | 8.7 | 1.9 | 2.9 | -4.5 | -6.0 | 9.4 | 10.5 | 11.8 |
| | SSP3 | 13.2 | 2.2 | 24.8 | 12.7 | 9.1 | 10.2 | 10.5 | 11.7 |
| | SSP5 | 11.2 | 1.9 | 22.5 | 10.0 | 7.3 | 9.0 | 9.5 | 10.5 |
| MMM | SSP1 | 6.3 | 1.4 | -3.3 | -7.8 | -9.6 | 5.5 | 5.9 | 7.4 |
| | SSP3 | 7.0 | 1.4 | -1.2 | -7.9 | -10.8 | 5.7 | 5.8 | 7.2 |
| | SSP5 | 7.2 | 1.4 | 3.2 | -4.0 | -6.7 | 4.2 | 4.5 | 5.8 |

**Table 2** Same analysis as in Table 1 for the far-future evaluation

| GCM | SSP | ΔP | ΔT | ΔQ1 | ΔQ2 | ΔQ3 | ΔE1 | ΔE2 | ΔE3 |
|---|---|---|---|---|---|---|---|---|---|
| GFDL | SSP1 | -4.5 | 1.3 | -34.8 | -34.0 | -36.0 | 0.3 | 0.3 | 2.0 |
| | SSP3 | -14.8 | 3.2 | -60.1 | -59.7 | -60.9 | -0.4 | -1.2 | -0.2 |
| | SSP5 | -17.2 | 3.7 | -65.2 | -64.2 | -65.1 | -0.3 | -1.8 | -1.1 |
| IPSL | SSP1 | 7.6 | 1.6 | -6.0 | -11.4 | -13.1 | 8.1 | 8.3 | 9.7 |
| | SSP3 | 18.4 | 4.3 | -9.4 | -11.0 | -11.6 | 22.9 | 22.5 | 23.7 |
| | SSP5 | 21.3 | 5.4 | -11.8 | -6.5 | -7.2 | 28.1 | 24.3 | 25.9 |
| MPI | SSP1 | -3.3 | 0.9 | -31.1 | -29.9 | -32.0 | 0.9 | 0.6 | 2.3 |
| | SSP3 | 8.3 | 1.9 | 4.9 | -3.0 | -4.7 | 1.6 | 2.8 | 4.0 |
| | SSP5 | 21.7 | 1.5 | 101.1 | 75.6 | 71.0 | -6.3 | -6.1 | -5.0 |
| MRI | SSP1 | -1.3 | 1.8 | -23.9 | -25.2 | -26.9 | 3.1 | 3.6 | 5.2 |
| | SSP3 | -2.7 | 3.6 | -33.5 | -33.1 | -33.8 | 5.7 | 5.5 | 6.7 |
| | SSP5 | -8.0 | 4.7 | -51.2 | -45.1 | -45.9 | 7.4 | 4.2 | 5.2 |
| UKESM | SSP1 | 7.4 | 2.0 | 7.3 | -1.6 | -4.3 | 7.7 | 8.2 | 9.6 |
| | SSP3 | 8.9 | 4.5 | 3.7 | -1.8 | -3.6 | 10.8 | 10.6 | 11.8 |
| | SSP5 | 17.8 | 5.8 | 19.1 | 14.8 | 12.1 | 17.3 | 15.6 | 16.9 |
| MMM | SSP1 | 1.2 | 1.5 | -17.7 | -20.4 | -22.4 | 4.0 | 4.2 | 5.8 |
| | SSP3 | 3.7 | 3.5 | -18.8 | -21.7 | -22.9 | 8.1 | 8.0 | 9.2 |
| | SSP5 | 7.2 | 4.2 | -1.6 | -5.1 | -7.0 | 9.2 | 7.2 | 8.4 |

**Literature**

Please note that literature already cited in the manuscript is not added to the following list.

Atkinson, P.M., Jeganathan, C., Dash, J., Atzberger, C., 2012. Inter-comparison of four models for smoothing satellite sensor time-series data to estimate vegetation phenology. Remote Sensing of Environment 123, 400–417. doi:10.1016/j.rse.2012.04.001

Hingerl, L., Bliefernicht, J., Guug, S., Sy, S., Neidl, F., Jagdhuber, T., Kunstmann, H., 2025. Comparative analysis of land–atmosphere interactions across three contrasting ecosystems in the west sudanian savanna. Journal of Hydrology: Regional Studies doi:10.1016/j.ejrh.2025.102751

Pianosi, F., Beven, K., Freer, J., Hall, J.W., Rougier, J., Stephenson, D.B., Wagener, T., 2016. Sensitivity analysis of environmental models: A systematic review with practical workflow. Environmental Modelling & Software 79, 214–232. doi:10.1016/j.envsoft.2016.02.008.

Saltelli, A., Ratto, M., Andres, T., Campolongo, F., Cariboni, J., Gatelli, D., Saisana, M., Tarantola, Stefano, 2008. Global sensitivity analysis: The primer. Wiley, Chichester, West Sussex.

Viovy, N., Arino, O., Belward, A.S., 1992. The best index slope extraction ( bise): A method for reducing noise in ndvi time-series. International Journal of Remote Sensing 13, 1585–1590. doi:10.1080/01431169208904212.

Weerasinghe, I., Bastiaanssen, W., Mul, M., Jia, L., and van Griensven, A.: Can we trust remote sensing evapotranspiration products over Africa?, Hydrol. Earth Syst. Sci., 24, 1565–1586, https://doi.org/10.5194/hess-24-1565-2020, 2020

Roland Yonaba, Angelbert Chabi Biaou, Mahamadou Koïta, Fowé Tazen, Lawani Adjadi Mounirou, Cheick Oumar Zouré, Pierre Queloz, Harouna Karambiri, Hamma Yacouba, A dynamic land use/land cover input helps in picturing the Sahelian paradox: Assessing variability and attribution of changes in surface runoff in a Sahelian watershed, Science of The Total Environment, Volume 757, 2021, 143792, ISSN 0048-9697, https://doi.org/10.1016/j.scitotenv.2020.143792

Zouré, C.O.; Kiema, A.; Yonaba, R.; Minoungou, B. Unravelling the Impacts of Climate Variability on Surface Runoff in the Mouhoun River Catchment (West Africa). Land 2023, 12, 2017. https://doi.org/10.3390/land12112017